# Comparing Reported Forest Biomass Gains and Losses in European and Global Datasets

**Lucas Sinclair** [1] and **Paul Rougieux** [2,*]

1  Sinclair.Bio—Bioinformatics and Data Science Consulting, 1212 Geneva, Switzerland; lucas@sinclair.bio
2  European Commission, Joint Research Center (JRC), Directorate D-Sustainable Resources, Bio-Economy, Via E. Fermi 2749, I-21027 Ispra, Italy
*  Correspondence: paul.rougieux@ec.europa.eu

**Abstract:** Net $CO_2$ emissions and sequestration from European forests are the result of removal and growth of flora. To arrive at aggregated measurements of these processes at a country's level, local observations of increments and harvest rates are up-scaled to national forest areas. Each country releases these statistics through their individual National Forest Inventory using their particular definitions and methodologies. In addition, five international processes deal with the harmonization and comparability of such forest datasets in Europe, namely the IPCC, SOEF, FAOSTAT, HPFFRE, FRA (definitions follow in the article). In this study, we retrieved living biomass dynamics from each of these sources for 27 European Union member states. To demonstrate the reproducibility of our method, we release an open source python package that allows for automated data retrieval and analysis, as new data becomes available. The comparison of the published values shows discrepancies in the magnitude of forest biomass changes for several countries. In some cases, the direction of these changes also differs between sources. The scarcity of the data provided, along with the low spatial resolution, forbids the creation or calibration of a pan-European forest dynamics model, which could ultimately be used to simulate future scenarios and support policy decisions. To attain these goals, an improvement in forest data availability and harmonization is needed.

**Keywords:** data harvesting; forest modeling; forest growth; macroecology; public data source comparison

## 1. Introduction

Forests consist of the largest terrestrial ecosystems that actively store carbon in living biomass. The sequestration effect is highly relevant in the context of climate change mitigation. Quantifying the magnitude of this effect remains a very active research topic [1]. In addition, forests are an important source of raw materials and renewable energy. For instance, substituting fossil-based materials with forest products provides additional climate mitigation benefits [2,3]. Also, as an integral ecosystem, forests provide countless benefits such as biodiversity habitats and recreation services.

In this context, researchers build models and run scenarios that simulate the synergies and trade-offs between these different demands that are made on forest ecosystems [4,5]. Scenarios and models are generally parametrized or trained to reproduce historical developments. As such, a prerequisite for the calibration of a model is to obtain precise and wide ranging data about the current and historical state of forests, as well as records of silvicultural practices. At the minimum these should include the explanatory variables of growth/yield and harvest models, i.e., forested area, species composition, increments and harvest rates.

National Forest Inventories (NFI) make data directly available for each European country individually through their websites. The direct use of NFI data would certainly offer a high level of detail and number of modeling features such as age breakdown and species composition. However, we were not able to use these data sources for the following reasons: (i) It would require parsing dozens of different websites which each have different

data formats and are written in different languages. (ii) The variables of interest have different definitions in each country and one does not have access to the process that links individual plot measurements to country-wide inventory data. For example differences in land use definitions might lead to incomparable carbon stock changes figures between NFI reporting and IPCC reporting. Before we move on to international datasets, the following paragraph mentions a few harmonization efforts.

The need to improve comparability has led NFIs to organize several harmonization projects, notably the COST action "usewood" [6] and the Distributed, Integrated and Harmonized Forest Information for Bioeconomy Outlooks (DIABOLO) project (one of the data sources mentioned below). Since 2012, the Joint Research Centre of the European Commission is participating in a "Framework Contract for the provision of forest data and services in support of the European Forest Data Centre" with the European network of NFIs (ENFIN). Until now, framework contracts have focused on the harmonization of forest area and biomass stock. Although it should be noted that the present study is not related to this framework contract and that it focuses on the dynamic aspects.

In the following analysis, we compared international data sources that offer information on forest biomass dynamics, on condition that the source in question includes a similarly formatted dataset for every one of the 27 EU member states. All the data acquired by these external sources originate at the individual NFIs, as they are the ones who prepare the values to fit inside the standardized reporting format of international organizations. Every country has developed its own methods to make their measurements conform to the questionnaires they receive. Typically, when there is a higher level of detail available at the national level, the data are summarized and filtered to fit the spreadsheet to be used. Conversely, when a lower level of detail is collected at the national level as compared to what is requested, data expansion is carried out through estimations [6]. To expand data, national correspondents can use regression techniques when data is available from case studies within the country or in other countries within the same biome. The data that we examine in our study are the outcome of such aggregation and interpolation processes performed by each country when preparing their reply to international surveys.

The following five public sources cover all 27 EU member states simultaneously and in a semi-unified format:

1.  IPCC: Intergovernmental Panel on Climate Change.
2.  SOEF: State of European Forests.
3.  FAOSTAT: Food and Agriculture Organization Statistics.
4.  HPFFRE: Harmonized projections of future forest resources in Europe.
5.  FRA: Forest Resource Assessment.

To acquire the data that these sources provide in an automated fashion, we built a software tool circumventing any missing bulk download functionality. Furthermore, we provided yet another level of standardization on top of what the international organizations do, reformatting and concatenating the measurements so as to enable the comparison of the different datasets amongst each other.

As to be expected, each source uses a different definition for what is considered to be forested land. This is caused, in part, by each organization being heavily shaped by different policy focuses. On top of differences in forest areas, each country has its particular way of measuring biomass dynamics. For instance, some sources report only on volumes of stems while others report only on masses of whole trees (including roots), hindering comparison.

Still, all forest information sources share the same fundamental principles. In essence, the state of today's forests is the result of past growth, natural mortality and disturbances. By definition, tree growth affects the stock of living biomass positively (gains) while natural mortality and disturbances affect it negatively (losses). Disturbances can be further distinguished in anthropogenic disturbances (harvest, atmospheric pollutants) and natural disturbances (storms, fire, pathogens) though losses data collected from the sources above

do not distinguish between the two. The distinction is complex because changes in natural mortality, anthropogenic and natural disturbances are frequently combined.

Changes in biomass volume through time can generally be described by the difference between increments (in green) and fellings (in red) visible in Figure 1. The gross increment box corresponds to all the above ground biomass growth. It was given a formal definition during the COST action usewood [6]. The net increment is the gross increment minus natural losses (Figure 1).

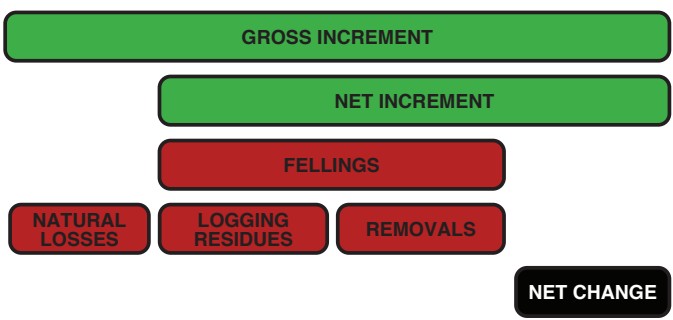

**Figure 1.** Schematic composition of increment and fellings reproduced from the "State of Europe Forests 2015" report [7].

By crossing these different datasets amongst each other, we were able to compare biomass growth and disturbances albeit at a very aggregated level. These aggregates should be useful for checking whether a forest dynamics model reproduces the historical biomass growth and disturbance trends or whether it diverges from official figures.

In contrast to forests in the boreal or tropical zones, a majority of Europe's forest are managed. Table A1 shows that most countries have close to 90% of their forest area available for wood supply according to the SOEF. Therefore it makes sense to compare the orders of magnitude of biomass dynamics across these sources even though they have different definitions of what constitutes forest land.

There is a need to facilitate exchange between scientific communities. For example, ref. [8] encourages collaboration between scientists contributing to the IPCC assessment reports and scientists contributing to the greenhouse gas inventory reports. In a similar vein, ref. [9] encourages data integration between sources to provide "detailed information at large spatial and long temporal scales that can be used in different modeling frameworks".

Finally, advances in remote sensing are changing the playing field. Forest dynamics modeling can only provide meaningful information when it is calibrated with ground data. Because of the high cost of data collection, inventory data are scarce with a periodicity of 5 to 10 years and a spatial resolution limited to a few hundred or a few thousand sample plots per country. Information gained from these sample plots is then extrapolated to the area of a whole country. In conjunction, airborne and space-based sensors complement ground data by providing repeated measurements over wide areas. However remote sensing observations from satellites and planes do not measure biomass directly. They measure various signals in the electromagnetic spectrum which are indicators of biomass stock (tree structure) or biological processes (photosynthesis). The temporal and spatial resolution of remote sensing methods has recently increased dramatically, with measurements available every year or even at a higher frequency and down to a spatial resolution of 30 m [10]. The detection of land use changes from forestry to agriculture is now performed routinely with a good accuracy at the global level. In permanently forested areas however, current remote sensing maps provide static information on the presence or absence of forest cover. Within a forested pixel, biomass has a fixed value [11]. When biomass is present, it is considered to be at a stable state without any notion of fluxes (input and outputs). Indeed, remote sensing based detection of tree growth or small scale biomass loss remains highly uncertain [12]. In the future, increased use of airborne LIDAR maps will help link field plot data to the scale of satellite based measurements [13].

In contrast to land use changes happening at the global scale, in EU countries, tree cover changes by less than 1% per year [7]. Therefore, the vast majority of forest biomass dynamics and related carbon fluxes happens in forest land remaining forest land. For the foreseeable future, ground measurement will remain the most precise source of information on slow growth processes. And this is where aggregating inventory data remains crucial.

## 2. Materials and Methods

### 2.1. Data Sources

We set out to retrieve data concerning forests on the European continent and to include only data sources that would provide measurements on multiple countries concurrently in a harmonized format. We focused on obtaining measurements expressing the amount of forested areas as well as growth and harvest rates.

There were several public data sources accessible online that provided these types of information in various forms and levels of granularity. Five data sources were identified and included in this study. Each one is detailed in the sections below.

We chose to include 27 countries in the analysis. Namely, all past and present European member states with the exception of Malta which has less than 400 hectares of forested land [7]. Not all countries were present in each data source, unfortunately.

### 2.1.1. IPCC

The "Intergovernmental Panel on Climate Change" (IPCC) is the United Nations body responsible for assessing the science related to climate change. An integral part of this process is the collection of emission estimates for each participating country. Under the name "National Inventory Submissions", they provide a common reporting format that assembles data on "all greenhouse gas (GHG) emissions and removals, implied emission factors and activity data" [14].

The IPCC provides a bulk download option [15] but it does not distinguish forest land emissions by losses and gains. To retrieve data from the IPCC National Inventory Submission website, we first parsed the HTML table [14] and selected the appropriate link for each country. For countries where several links were available, we picked the files that matched the country's ISO3 code. As the IPCC website currently blocks automated requests with the use of the "Incapsula" software solution, the pages and excel files were downloaded by running a headless browser with the "gecko-driver" and "selenium" [16] technologies to bypass their restrictions. As a side effect of the anti-DDoS techniques employed, their current interface is only practical for the manual retrieval of specific measurements. It actively prevents automated data retrieval and therefore impairs scientists from performing wide-ranging analyses.

We then developed routines that parsed the resulting excel sheets. We were interested in the carbon stock changes in forest biomass reported in "Table 4.a Sectoral background data for land use, land-use change and forestry—forest land" and focused only on forest land that remained as forest land. From this table, we used the columns titled "Carbon stock change in living biomass" subdivided in gains, losses and net change.

Countries varied in the structure of the aforementioned table and the number of rows provided. For instance, some countries used the different rows to distinguish coniferous and deciduous forest land while other countries had no such rows. Other times, the additional rows were used to distinguish between categories such as mainland versus overseas territories instead of forest types. This is due to the fact that the IPCC common reporting format does not impose a specific subdivision of the forest land category, and many countries add or remove rows at will, specifying custom information or not. This often prevents automated parsing of all countries. Fortunately, the code automatically identifies the length of the contained table. However it may require some manual intervention to adapt the software to a new country's data structure as it becomes available.

### 2.1.2. SOEF

The "Ministerial Conference on the Protection of Forests in Europe" [7] regularly publishes a "State of European Forests Report" (SOEF). This report includes an aggregation of NFI data that is made possible by a common reporting standard.

Each countries' submissions are accessible in a database [17]. To retrieve the data concerning all countries, we parsed the HTML contents of the drop-down menu and automated the download of all excel files based on the links extracted. Following the download, every country's data were contained in a single Excel file. For each file, we parsed four tables:

- Table 1.1a: Forest area.
- Table 1.1b: Forest area by forest types.
- Table 1.2b: Growing stock by forest types.
- Table 3.1: Increment and fellings.

This was enabled by developing a flexible table parser that can identify the start and end rows within each different file to streamline the process and avoid manual interventions. This was necessary as the excel files differed greatly between countries and do not seem to be post-processed or corrected by SOEF themselves. This is made evident by the presence of calculations and temporary notes written in local languages next to the tables (in usually empty cells), as well as other typos, mistakes and inconsistencies. To help with ease of access and comparison, column titles were renamed and units were converted to SI standards where possible.

The main table of interest in the SOEF source contains the increment and fellings data. There are potential differences in land use definitions and in stock definitions between the SOEF and IPCC. We didn't account for the differences in forest land definitions, but we did account for the differences in stock definitions. Further explanations on the conversion from volumes of tree stem (over bark) to tons of carbon are available in the section on conversion to mass. To choose conversion factors, we used the tables on forest area by forest types and growing stock by forest types to compute a stock per forest type per hectare.

### 2.1.3. FAOSTAT

FAOSTAT is the corporate statistical database of the "Food and Agriculture Organization of the United Nations". The `forest_puller` software downloads forest area [18] and wood removals data [19]. In the case of the "Forest Land" dataset, we filtered the data by picking rows where "element" was equal to "area" and where "item" was equal to "forest land". In the case of the "Forestry Production and Trade" dataset, wood removals were determined by picking the rows where "element" was equal to "production quantity". Furthermore, we selected all the "roundwood" and "wood fuel" items, whether they were coniferous or non-coniferous. The wood removals obtained here are narrower in scope that the IPCC loss data, since they cover only the productive forest land. They are expressed in cubic meters under bark, we converted them to tons of carbon using the methodology below.

### 2.1.4. FRA

FRA stands for "Forest Resource Assessment" and is a report that is published every five years by the FAO (same organization as FAOSTAT). This report provides global information on forest area, stock and additional sustainability indicators.

The software produced downloads two datasets from the "CountrySTAT" platform. The first is titled "Extent of forest and other wooded land" [20]. We filtered this dataset by selecting rows where "category" was equal to "forest" to obtain the total area for each country. The second is titled "Growing stock by forest/other wooded land" [21]. We filtered this dataset by selecting rows where "category" was equal to "total growing stock" and "land type" was equal to "forest" to retrieve the standing stock in each country. This source doesn't provide data on the dynamics, i.e., increments and fellings, therefore we only report the forest area in the results.

### 2.1.5. HPFFRE

HPFFRE stands for "Harmonized projections of future forest resources in Europe" [22]. The supplementary data [23] covers 21 of the 27 countries studied in this analysis. It was released as part of a work package in the Diabolo project [24].

We selected forest area and felling volumes from the first scenario and used the historical period only, discarding all future predictions made by the model. We also summed the different categories of availability for wood supply together (FAWS, FNAWS, FRAWS).

### 2.2. Conversion to Mass

Though all data sources describe the same phenomena of tree growth and removal, not all use the same definitions or units. The IPCC data source provides measures of carbon stock change in the living biomass using tonnes of carbon per hectare as units. In their case, the term biomass includes the tree stem, the branches and the roots. The same unit is used for both increments and removals.

Meanwhile, SOEF provides increment and felling volumes using cubic meters per hectare and including only the stem of the tree (over bark). FAOSTAT describes roundwood removals measurement as "all quantities of wood felled and removed from the forest and other wooded land or other felling sites. They are measured in cubic meters under bark (without bark)". Lastly, HPFFRE expresses the dynamics in "Stemwood volume measured over bark expressed as unit area volume". It further specifies: "Total stemwood volume measured over bark. Part of tree stem from the felling cut to the tree top with the branches removed, including bark". A summary of the different units used is provided in Table 1.

**Table 1.** Overview of the variables of interest in each dataset.

| Source | Area | Stock | Gains | Losses | Units |
|--------|------|-------|-------|--------|-------|
| IPCC | x | x | x | x | Biomass in tonnes of carbon |
| SOEF | x | x | x | x | Stem volume in m$^3$ over bark |
| FAOSTAT | x | | | x | Stem volume in m$^3$ under bark |
| FRA | x | x | | | Stem volume in m$^3$ over bark |
| HPFFRE | x | x | | x | Stem volume in m$^3$ over bark |

As three of the sources (SOEF, FAOSTAT and HPFFRE) report forest dynamics in volume of stem wood (over bark or under bark) while IPCC reports biomass dynamics in tonnes of carbon, for the purposes of comparison, we converted the merchantable volume increment to a carbon biomass gain (including both above and below ground biomass) using Equation (1) based on the IPCC guidelines [25]:

$$G_{mc} = I_v \times BCEF_I \times (1 + R) \times CF \qquad (1)$$

where:

- $G_{mc}$ is the carbon biomass gain [kg/ha/year].
- $I_v$ is the merchantable volume increment [m$^3$/ha/year].
- $BCEF_I$ is the biomass conversion and expansion factor of the annual increment; it accounts for both the density and the expansion of merchantable biomass to above ground biomass [kg/m$^3$].
- $R$ is the root to shoot ratio or the "ratio of below-ground biomass to above-ground biomass (r)" [25] [unitless].
- $CF$ is the carbon fraction of dry biomass [unitless].

Similarly, we converted wood removal volumes $H_v$ to losses in tonnes of carbon $L_{mc}$ according to Equation (2) based on [25]:

$$L_{mc} = H_v \times BCEF_R \times (1 + R) \times CF \qquad (2)$$

where:

- $G_{mc}$ is the carbon biomass loss [kg/ha/year].
- $H_v$ is the merchantable volume harvest [m$^3$/ha/year].
- $BCEF_R$ is the expansion factor of wood and fuelwood removal volume to aboveground biomass removal [kg/m$^3$].
- $R$ and $CF$ are the same as in Equation (1).

The 2006 IPCC guidelines [25] distinguish the $BCEF$ and $R$ parameters along different criteria. First, the criteria to choose $BCEF_I$, $BCEF_R$ and $BCEF_S$ are: climatic zone, forest type and growing stock level in cubic meters. We allocated each country to one or two climatic zones based on the FAO map of global ecological zones [26]. We used the SOEF forest stock data to compute a merchantable biomass stock per hectare distinguished by coniferous and broadleaves. Based on that stock value, we choose separate BCEF parameters for coniferous and broadleaves in each country. Since country level gains and losses are not distinguished by forest types, we combined each BCEF into a weighted average, using the proportion of coniferous and broadleaved forest stock as a weighting factor. Second, the criteria to choose $R$ are: domain, ecological zone, forest type and a threshold expressed in tons of aboveground biomass per hectare. We multiplied the merchantable stock by $BCEF_S$ to obtain the above ground biomass stock and used it as a threshold value (along climatic zone and forest type) to choose the $R$ parameter. Since gains and losses are not distinguished by forest types, we used the ratio of conifer and broadleaves in the stock to obtain a single weighted average value of $R$ for each country. For countries belonging to 2 climatic zones, the R values of the 2 climate zones are used in the weighed mean.

Based on the IPCC guidelines [25], we selected a single value for the carbon fraction of dry matter $CF = 0.47$. This value expresses tonnes of carbon per tonne of dry biomass.

In order to convert the under-bark volumes reported by FAOSTAT to over-bark volumes, we used an average value of "Volume ratio wood/bark plus wood" equal to 0.88 [27].

Several simplifying assumptions were made when choosing $BCEF$ and $R$. For countries belonging to 2 climatic zones, the forest area was split equally between 2 climatic zones and the proportion of coniferous and broadleaves were considered to be the same in the 2 climatic zones. Different definitions of forested land use, mean that the biomass changes per hectare do not refer to the same areas. For example some data sources are limited to forest land available for wood supply while other sources include forest land not available for wood supply, even so, FAWS covers more than 90% of the total forest area in most countries according to the SOEF (Table A1). Increments and removals already have an unknown level of uncertainty in the input data and the conversion to tons of carbon increases the uncertainty. Therefore when using these values for comparison purposes, we should look only at their orders of magnitude and keep in mind that there is a large margin of error.

### 2.3. Common Data Format

From a software implementation perspective, the parsing of FAOSTAT bulk data is the most reproducible through time and across countries. The experience FAOSTAT has acquired by providing publicly available agricultural data for decades is apparent as the data are more standardised. Other sources tend to have ill-formtted spreadsheet for some countries or some years. We ended up implementing special conditions in the data preparation software to work around non-standard data input. It is likely that more of these special conditions will have to be implemented in future updates, rendering the updated process less automated than we would have desired.

Following the procedure described above for each data source, the results were compiled in a common format for all countries by using the `pandas` python package [28] and its `DataFrame` object. Data frames enable grouped computations, pivoting and table joins necessary to transform the variables to comparable units. Pandas data frames are also compatible with the `matplotlib` library [29] used for the graphs and visualizations.

### 2.4. Open Software

All the software written for the purpose of this publication to download, store, process, parse and visualize the forest data is freely available as a python package under the name `forest_puller`. The package can be installed with the command `pip3 install forest_puller` and the source code is distributed online [30] under the MIT license. For example Equations (1) and (2) are expressed in a vectorized form in the source code of `forest_puller/viz/converted_to_tons.py` line 85 and 87 of commit 6d713d8. We would like to encourage readers to review and contribute to the software as well as report any issues encountered on the bug tracker. Each source file has ample comments and every routine is documented.

## 3. Results and Discussion

### 3.1. Comparison of Forest Area

The main goal was to compare the changes in biomass volumes. Since these changes are always normalized by the area, it is natural to start comparing forest areas first (Figure 2). For SOEF, HPFFRE and FRA, the total forest area excludes other wooded land, i.e., land not defined as forest, though it should be noted that forest infrastructure such as roads and ditch networks can be reported under forest land. The maximum forest area is shown for each country in Tables A2 and A3 compares the forest area for the most recent years available in each source. In most countries, the total forest area reported by FAOSTAT is identical to the one reported by SOEF. The former has a periodicity of one year while the latter has a periodicity of five years. Additional points in FAOSTAT's yearly data have been obtained by interpolation as is visible in the changes of slope for Denmark and Bulgaria, for example. As the dynamics reported by SOEF and FAOSTAT are highly similar, we will focus on the comparison between the IPCC and SOEF forest areas for the rest of this section.

In Figure 2, four types of patterns emerged: (i) countries for which IPCC and SOEF forest areas are identical: Czechia, Hungary, Ireland, Italy, Sweden, (ii) counties for which the trends are similar but the curves are separated by an offset which could be due to a different forest land definition: Austria, Belgium, Croatia, Cyprus, Germany, Luxembourg, Slovakia, United Kingdom, (iii) countries for which the trends differ only slightly: Bulgaria, Estonia, France (footnote Guyana), Lithuania, Luxembourg, Poland, Portugal, Slovenia and Spain, (iv) countries for which the trends differ markedly: Denmark, Finland, Greece, Latvia, Lithuania, The Netherlands, Romania.

Forest dynamics are modeled differently between productive and non-productive forest land. This distinction is made available in the HPFFRE dataset which reports (i) forest available for wood supply, (ii) forest with restricted availability for wood supply, and (iii) forest not available for wood supply. However, in the IPCC data, this distinction is not available for all countries.

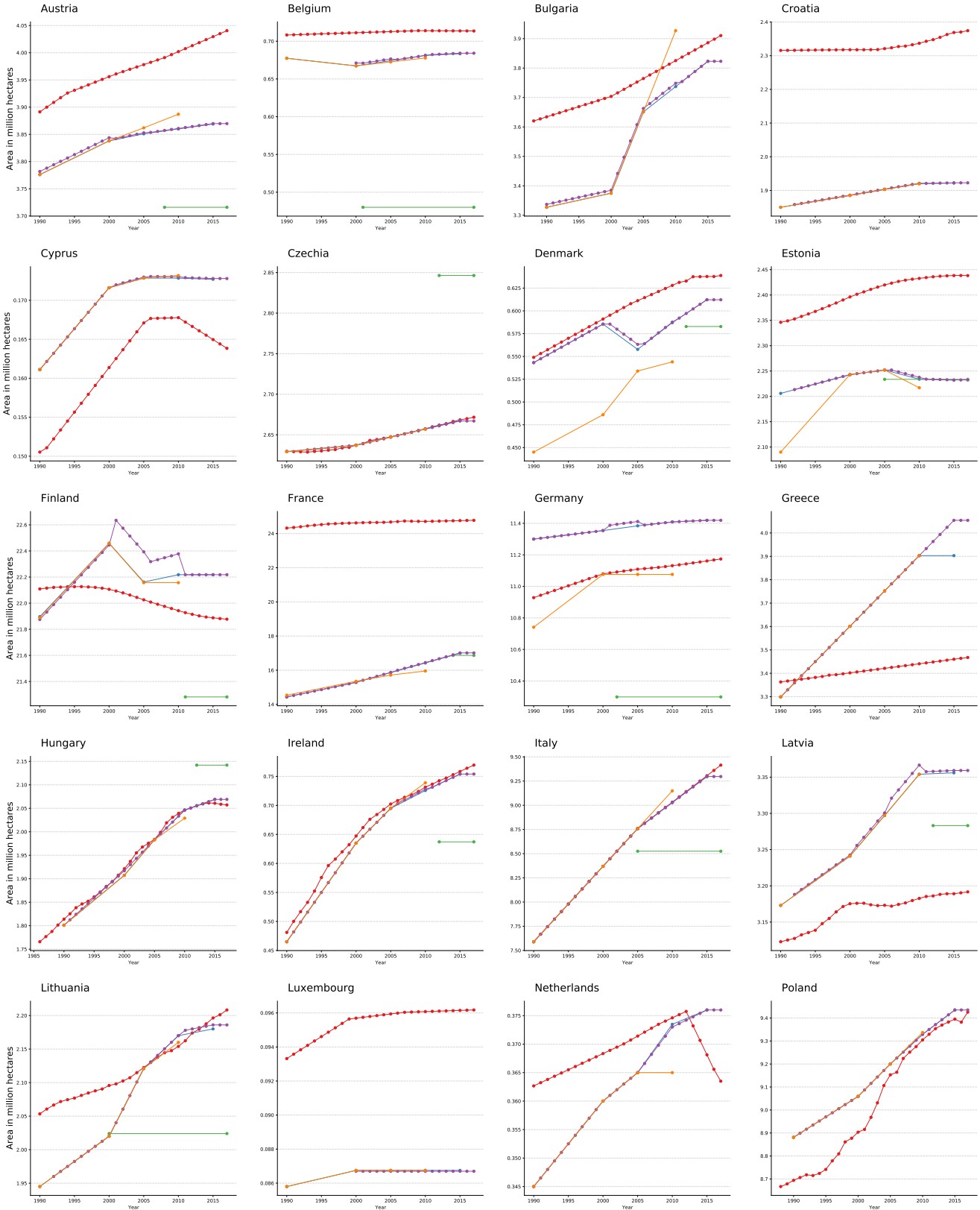

**Figure 2.** *Cont.*

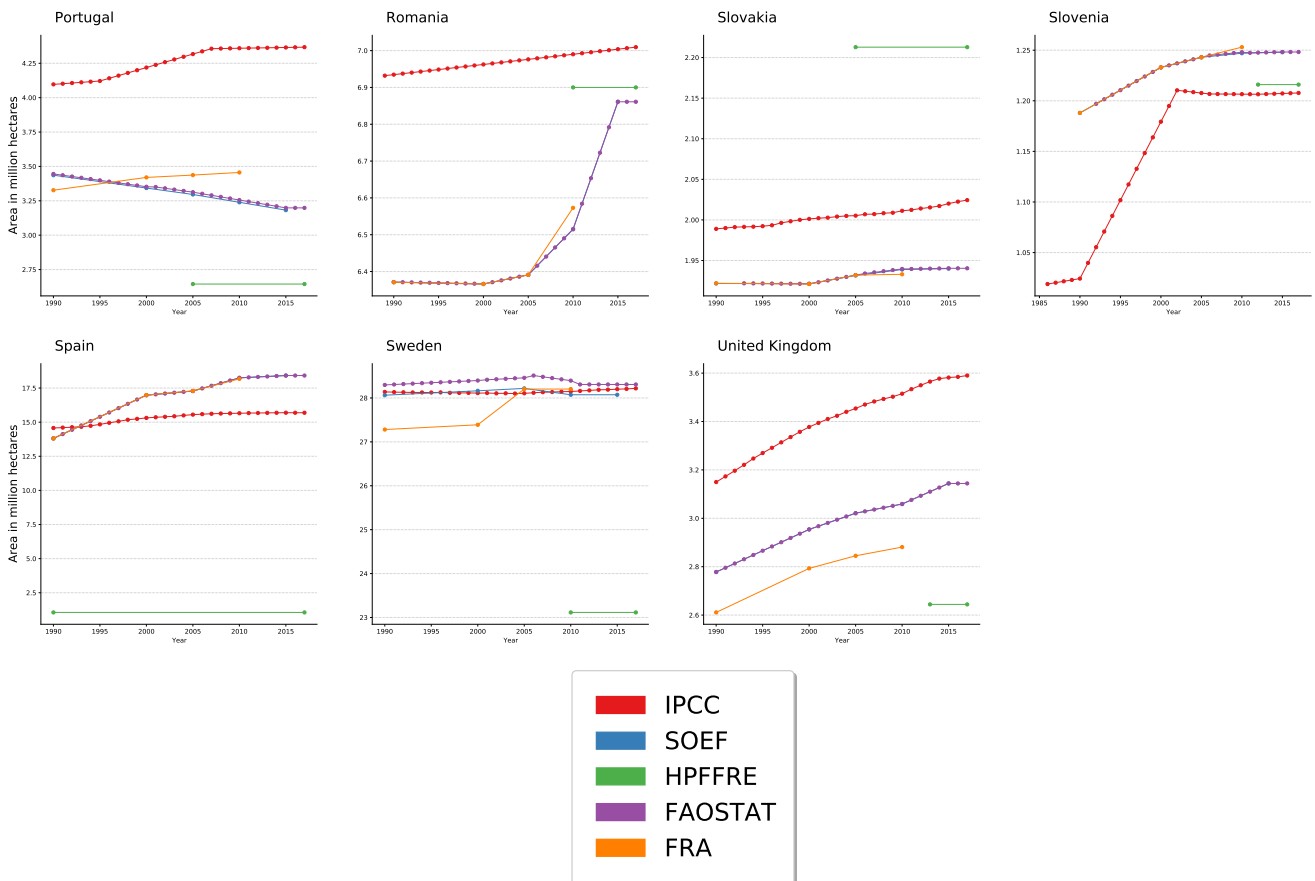

**Figure 2.** Total forest area in million hectares in the five data sources and 27 countries present in the dataset.

### 3.2. Growth Dynamics

International processes provide methods to aggregate biomass gains into comparable figures. However, conversion factors were not available for all countries, hence Figures 3 and A1 are expressed in the original units. In order to compare other sources with the IPCC, which expresses gains in tonnes of carbon for the total above and below ground biomass, we converted all sources to tonnes of carbon (see Figures 4 and A2).

In most countries, biomass gains are stable or slightly increasing over the period. Net increment values are around five to six cubic meters per hectare according to the SOEF data. Corresponding gain values are below two tonnes of carbon per hectare in the IPCC data. The following countries show similar stable or slightly increasing trends both in terms of IPCC gains and SOEF increment values: Austria, Croatia, Finland, France, Italy, Poland, Romania, Hungary. However, trends differ for some countries. Indeed, the IPCC gain values are slightly decreasing year over year, while the SOEF data shows increases in gains over the same period of time in Denmark, Netherlands, Slovenia and Lithuania. In the case of Latvia, IPCC gain values have a decreasing trend over the period while the reported SOEF data are stable. Other countries do not have enough data points to compare trends.

Germany and Belgium use the stock change approach to report biomass gain values to the IPCC. As a result, the green curve Figure A1 has typical step shapes with constant gains for a few years followed by large changes. Other countries use the approach "one inventory plus change" which causes the curve to have more gradual annual changes along a trend.

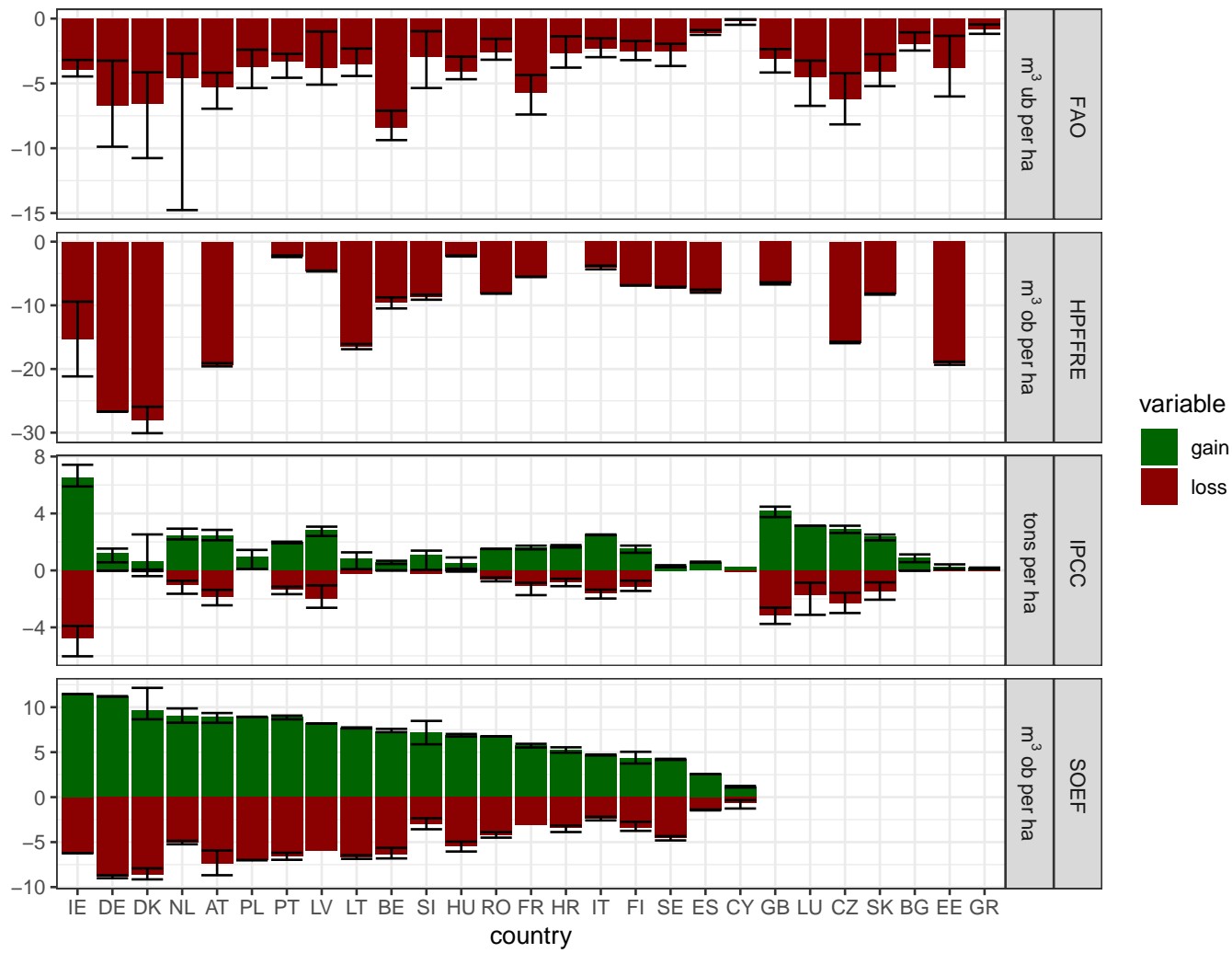

**Figure 3.** Average gains and losses by country for each source. Error bars represent the minimum and maximum values of the time series.

Figure 4 compares biomass growth between 1990 and 2015, for a selection of countries (Austria, Croatia, Finland, France, Italy, Romania). Plots for all countries are available in the Appendix A, Figure A2. The following countries have similar biomass gain levels in the IPCC and SOEF after conversion to tonnes of carbon: Austria, Croatia, Cyprus, Finland, France, Latvia, The Netherlands. Another comparison provided here are the gains averaged over the whole period available in Table A3 and Figure 3. We can see that the IPCC gain level is higher than the corresponding SOEF gain in the case of Italy. Conversely, the IPCC gain level is lower than the one found in SOEF for Denmark, Romania, Slovenia.

A lower gain value is expected in countries where the IPCC forest area is larger than the SOEF forest area. Since the larger area is likely to include more of the unproductive forest land possessing slow growth rates, this lowers the average growth value.

A comparison in tonnes of carbon is not possible in other countries due to a lack of data in one of the sources or a missing conversion factor. Even though the conversion from stem volume increment to biomass gain is approximate, the fact that seven countries have similar values seemed to confirm that the approach is relevant to check the order of magnitude of biomass gains at a national level.

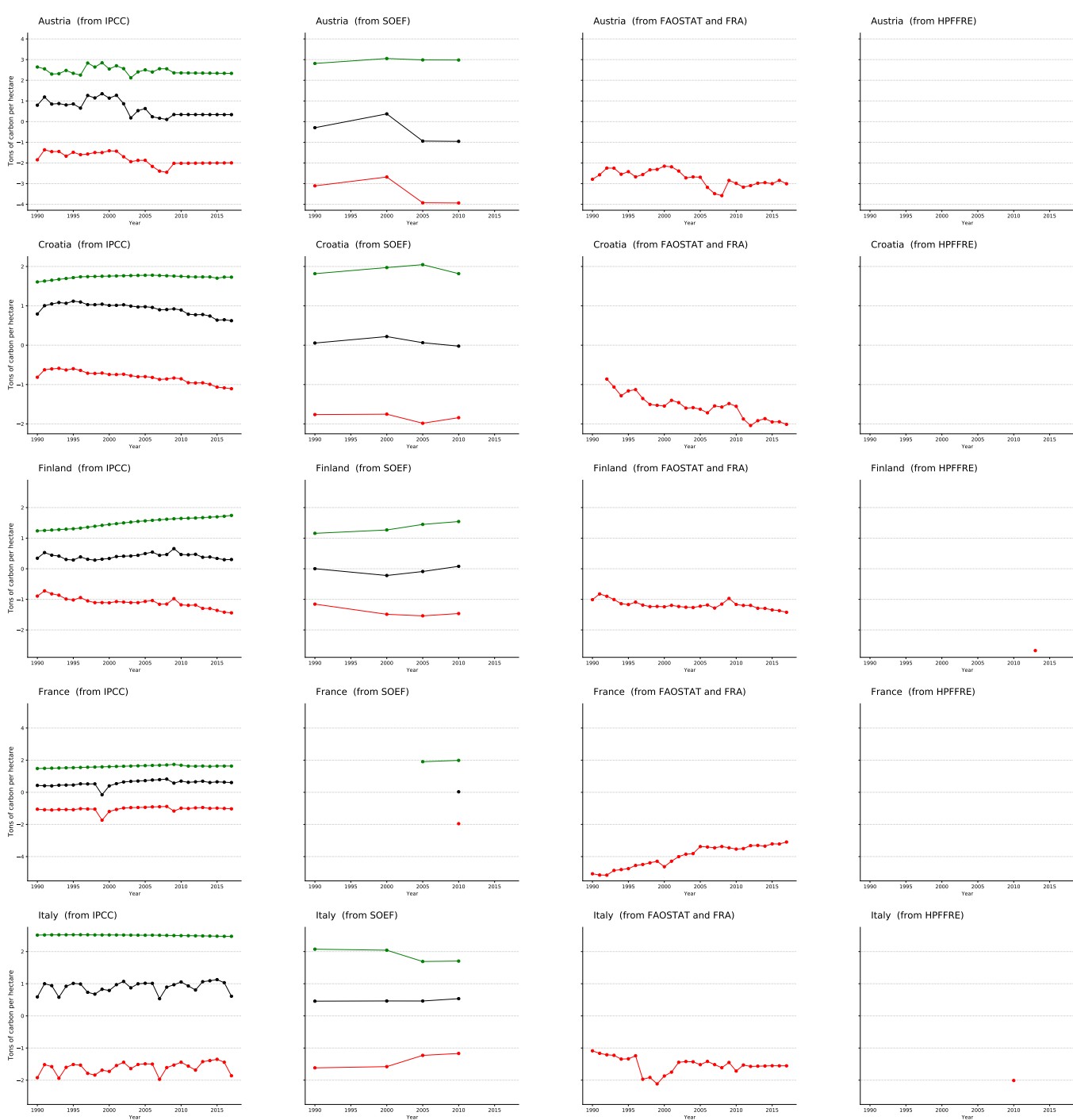

**Figure 4.** *Cont.*

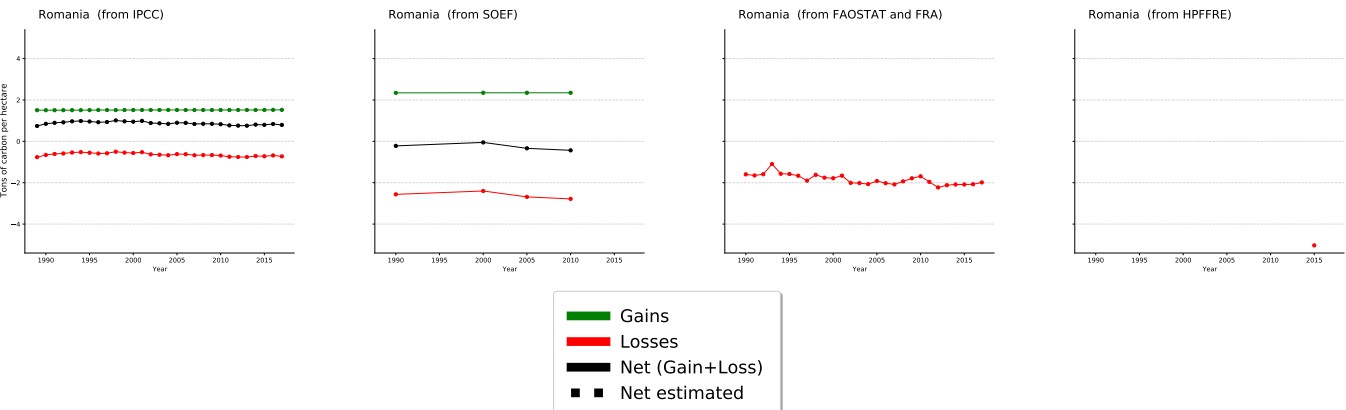

**Figure 4.** Comparison of the time series of forest dynamics for a selection of countries. All sources are expressed in the same unit (tonnes of carbon).

### 3.3. Disturbances Dynamics

Biomass losses are due to the combination of harvest and natural disturbances. At the national level, changes in losses can be due to large fluctuations in economic activity. As an example, in many countries, the impact of the 2008 financial crisis can be observed by a decrease in harvest. Indeed, the red curve moves upward as losses are represented by negative values on the vertical axis (Figure 4). Then both types of disturbances can be combined. For instance, large storms or insect outbreaks lead to significant amounts of salvage logging visible in the national harvest statistics. A striking illustration is visible in the losses curves of Austria and Czechia as both countries have been severely hit by storms in recent years. Beyond the issue of combined disturbances, one should remain cautious and remember that in the international sources analysed in this article, disturbances losses are reported without any indication of the associated uncertainty. Improved methods to indicate the uncertainty of the estimates would help in understanding the significance of changes in the time series.

In the following countries, biomass loss levels were lower (in absolute value) in IPCC than in SOEF: Austria, Croatia, Cyprus, Denmark, Finland, Latvia, Netherlands, Romania, Slovenia. Countries where biomass loss levels were higher (in absolute value) in IPCC than in SOEF were: Ireland and Italy.

Considering the stark differences of reported loss values (Table A3 and figure 3), there is likely an issue with the expansion factors used in Equation (2). Further analysis of the reasons for these discrepancies would require a more detailed model benefiting from growing stock level broken down by species and climate zones.

On a planetary scale, taking into account indirect land use effect is crucial to avoid underestimating forest emissions, i.e., overestimating the forest sink effect [8]. Additionally, anthropogenic carbon missions lead to increases in natural disturbances and mortality. However, these effects are not separable from the base line disturbances and natural mortality in the National Inventory Reports data. We also note that future climate change is likely to continue impacting the interaction between disturbance agents [31].

### 4. Conclusions

The discrepancies across the five sources varied greatly from one country to another. It is difficult to identify the reasons for this incoherence as many sources of errors are compounded in national aggregates. More specifically, the variability along tree species composition, age distribution, soil and climatic conditions is lost in the aggregation process. Disaggregating data across these variables would help to find out the reasons for the differences observed.

Fundamentally, the raw measurements on which all estimates are based, are obtained by observing forests on a very small set of areas localized in space and time. The

country-wide forest growth rates are but aggressive interpolations made from these ground measurements. Furthermore, these essential plot data obtained from field campaigns are habitually not divulged by the NFIs as they are protected by statistical confidentiality. To expand the spatial and temporal scales, more players would need to release harmonised public information as close as possible to these primordial quantifications, while continuing to ensure statistical confidentiality.

The data harvesting and merging software we introduce in this article can be reused by others. All data conversion steps have been developed in the python programming language. The software should be capable of updating data automatically as new data becomes available. Though future changes in the structure of the input data might require slight adjustments to the code.

Grassi et al. [8] call for the global vegetation modeling community to: "[...] design future models and model experiments to increase their comparability with historical [Green House Gas Inventories] and thus their relevance in the context of the Paris Agreement". We hope the software module we produced can provide an overview of the biomass losses and gains at national levels and can facilitate comparison attempts by the vegetation and carbon cycle modeling community in the future. The harmonized data assembled here are not sufficient to calibrate a European forest dynamics model, but it provides a series of reference points necessary to validate such a model on historical data. The underlying software demonstrates how to structure the data acquisition and how to implement a conversion algorithm. Can it be a meaningful step towards "the availability and provision of harmonized freely-available databases" [9]?

In the future, the spatial and temporal precision of remote sensing data will continue to increase and maps of biomass change will become available. There will be a pressing need to compare them with ground based observations of biomass losses and gains. At the international level, these comparisons can be supported by a framework for sharing ground based observation. Building such a framework will be very challenging. It will be challenging on the scientific level because each ground data collection is adapted to its own biome and it will be challenging on the policy level because each national forest inventory effort is shaped to its particular socio-economic context.

**Author Contributions:** Conceptualization, L.S. and P.R.; Data curation, L.S. and P.R.; Formal analysis, L.S. and P.R.; Methodology, P.R.; Software, L.S.; Visualization, L.S.; Writing–original draft, L.S. and P.R. All authors have read and agreed to the published version of the manuscript.

**Funding:** This research received no external funding.

**Institutional Review Board Statement:** Not applicable.

**Informed Consent Statement:** Not applicable.

**Data Availability Statement:** The data is accessible through the python package `forest_puller` [30].

**Acknowledgments:** We would like to thank our colleagues of the JRC biomass team. In particular, Roberto Pilli and Raul Abad Vinas for introducing us to the IPCC inventory submission dataset, Valerio Avitabile, Andrea Camia, Sarah Mubareka and Nicolas Robert for comments on an early version of this article and Bernd Eckhardt for sharing his experience concerning NFIs data platforms. The work described in this paper has (though not constituting its official output) been carried out in the context of the JRC Biomass assessment study https://ec.europa.eu/knowledge4policy/projects-activities/jrc-biomass-study_en.

**Conflicts of Interest:** The authors declare no conflict of interest. The funders had no role in the design of the study; in the collection, analyses, or interpretation of data; in the writing of the manuscript, or in the decision to publish the results.

## Abbreviations

The following abbreviations are used in this manuscript:

| | |
|---|---|
| API | Application Programming Interface |
| AWS | Available for wood supply |
| CRF | Common Reporting Format |
| CSV | Comma Separated Values format |
| DDoS | Distributed Denial of Service |
| EU | European Union |
| FAOSTAT | Food and Agriculture Organization Statistics |
| FAWS | Forest available for wood supply |
| FNAWS | Forest not available for wood supply |
| FRAWS | Forest available for wood supply with restrictions |
| GHG | Green House Gas |
| GUI | Graphical User Interface |
| HTML | Hypertext markup language |
| HPFFRE | Harmonized Projections of Future Forest Resources in Europe |
| IPCC | Intergovernmental Panel on Climate Change |
| ISO | International Standardization Organization |
| MIT | Massachusetts Institute of Technology |
| SI | The International System of Units |
| SOEF | State Of European Forests |
| XLS | Microsoft Excel Spreadsheet format |
| ZIP | Compressed file archive format |

## Appendix A. Extra Tables and Graphs

**Table A1.** Proportion of area Available for Wood Supply (AWS) and Forests with Restrictions on Availability for Wood Supply (FRAWS) in the total forest area based on data from SOEF and HPFFRE.

| Country | SOEF | HPFFRE | |
|---|---|---|---|
| | AWS | AWS | AWS + FRAWS |
| AT | 86.3% | 85.4% | 94.3% |
| BE | 98.1% | 100.0% | - |
| BG | 57.9% | - | - |
| CY | 23.8% | - | - |
| CZ | 86.3% | 95.0% | - |
| DE | 95.3% | 95.5% | 99.2% |
| DK | 93.5% | 96.2% | - |
| EE | 89.3% | 77.3% | 90.3% |
| ES | 79.9% | 94.7% | - |
| FI | 87.6% | 79.3% | 89.9% |
| FR | 94.3% | 76.4% | 94.7% |
| GB | 100.0% | 100.0% | - |
| GR | 92.1% | - | - |
| HR | 90.5% | - | - |
| HU | 86.0% | 96.8% | - |
| IE | 83.8% | 83.8% | 99.4% |
| IT | 88.4% | 93.8% | - |
| LT | 88.3% | 87.1% | 98.8% |
| LU | 99.3% | - | - |
| LV | 93.9% | 97.1% | - |
| NL | 80.1% | - | - |
| PL | 87.3% | - | - |
| PT | 65.6% | 59.3% | - |

**Table A1.** *Cont.*

| Country | SOEF | HPFFRE | |
|---|---|---|---|
| | AWS | AWS | AWS + FRAWS |
| RO | 67.4% | - | - |
| SE | 70.6% | 96.2% | - |
| SI | 91.3% | 90.0% | - |
| SK | 92.0% | 94.9% | 98.0% |

**Table A2.** Maximum forest area in million of hectares (at any year) for each country by sources.

| Country | IPCC | SOEF | FAOSTAT | HPFFRE | FRA |
|---|---|---|---|---|---|
| AT | 4.040 | 3.869 | 3.870 | 3.716 | 3.887 |
| BE | 0.714 | 0.683 | 0.684 | 0.480 | 0.678 |
| BG | 3.910 | 3.823 | 3.823 | - | 3.927 |
| HR | 2.374 | 1.922 | 1.923 | - | 1.920 |
| CY | 0.168 | 0.173 | 0.173 | - | 0.173 |
| CZ | 2.672 | 2.667 | 2.667 | 2.846 | 2.657 |
| DK | 0.639 | 0.612 | 0.612 | 0.583 | 0.544 |
| EE | 2.438 | 2.252 | 2.252 | 2.234 | 2.252 |
| FI | 22.127 | 22.459 | 22.635 | 21.282 | 22.459 |
| FR | 24.775 | 16.989 | 17.013 | 16.866 | 15.954 |
| DE | 11.174 | 11.419 | 11.419 | 10.299 | 11.076 |
| GR | 3.468 | 3.903 | 4.054 | - | 3.903 |
| HU | 2.061 | 2.069 | 2.069 | 2.142 | 2.029 |
| IE | 0.769 | 0.754 | 0.754 | 0.637 | 0.739 |
| IT | 9.415 | 9.297 | 9.297 | 8.525 | 9.149 |
| LV | 3.192 | 3.356 | 3.367 | 3.283 | 3.354 |
| LT | 2.208 | 2.180 | 2.186 | 2.024 | 2.160 |
| LU | 0.096 | 0.087 | 0.087 | - | 0.087 |
| NL | 0.376 | 0.376 | 0.376 | - | 0.365 |
| PL | 9.426 | 9.435 | 9.435 | - | 9.337 |
| PT | 4.367 | 3.436 | 3.445 | 2.645 | 3.456 |
| RO | 7.009 | 6.861 | 6.861 | 6.900 | 6.573 |
| SK | 2.024 | 1.940 | 1.940 | 2.213 | 1.933 |
| SI | 1.210 | 1.248 | 1.248 | 1.216 | 1.253 |
| ES | 15.694 | 18.418 | 18.418 | 1.057 | 18.173 |
| SE | 28.218 | 28.218 | 28.511 | 23.115 | 28.203 |
| GB | 3.590 | 3.144 | 3.144 | 2.644 | 2.881 |

**Table A3.** Average gains and losses in tons of carbon per hectare over the data period available.

| Source | Gains per Hectare | | | | Losses per Hectare | |
|---|---|---|---|---|---|---|
| | IPCC | SOEF | FAO | HPFFRE | IPCC | SOEF |
| **Country** | | | | | | |
| AT | 2.45 | 2.96 | −2.74 | - | −1.81 | −3.41 |
| BE | 0.56 | 2.24 | −4.60 | −4.23 | −0.00 | -3.05 |
| BG | 0.89 | - | −1.33 | - | −0.02 | - |
| CY | 0.24 | 0.31 | −0.09 | - | −0.11 | −0.26 |
| CZ | 2.90 | - | −3.18 | - | −2.27 | - |
| DE | 1.18 | - | - | - | −0.02 | - |
| DK | 0.64 | 3.04 | −3.94 | - | −0.11 | −4.66 |

**Table A3.** *Cont.*

| Source | Gains per Hectare | | | Losses per Hectare | | |
|---|---|---|---|---|---|---|
| | **IPCC** | **SOEF** | **FAO** | **HPFFRE** | **IPCC** | **SOEF** |
| **Country** | | | | | | |
| EE | 0.27 | - | −2.41 | −10.85 | −0.00 | - |
| ES | 0.57 | 1.05 | −0.82 | - | - | −0.96 |
| FI | 1.50 | 1.35 | −1.18 | −2.67 | −1.10 | −1.41 |
| FR | 1.61 | 1.95 | −3.99 | - | −1.04 | −1.95 |
| GB | 4.19 | - | −1.96 | −3.47 | −3.16 | - |
| GR | 0.16 | - | - | - | −0.01 | - |
| HR | 1.73 | 1.91 | −1.56 | - | −0.81 | −1.83 |
| HU | 0.49 | 2.39 | −3.03 | - | −0.02 | −3.62 |
| IE | 6.52 | 4.16 | −2.59 | - | −4.76 | −3.66 |
| IT | 2.51 | 1.88 | −1.52 | −2.01 | −1.61 | −1.40 |
| LT | 0.84 | 2.86 | −2.47 | - | −0.20 | −4.04 |
| LU | 3.14 | - | - | - | −1.69 | - |
| LV | 2.81 | 2.73 | −2.47 | - | −1.98 | −3.40 |
| NL | 2.45 | 3.05 | −2.97 | - | −1.03 | −2.87 |
| PL | 0.96 | 2.88 | −1.93 | - | - | −3.20 |
| PT | 1.99 | - | - | - | −1.32 | - |
| RO | 1.51 | 2.35 | −1.84 | −5.03 | −0.64 | −2.61 |
| SE | 0.33 | 1.32 | −1.31 | - | - | −2.09 |
| SI | 1.09 | 2.43 | −1.51 | - | −0.23 | −1.36 |
| SK | 2.35 | - | −2.27 | −3.97 | −1.47 | - |

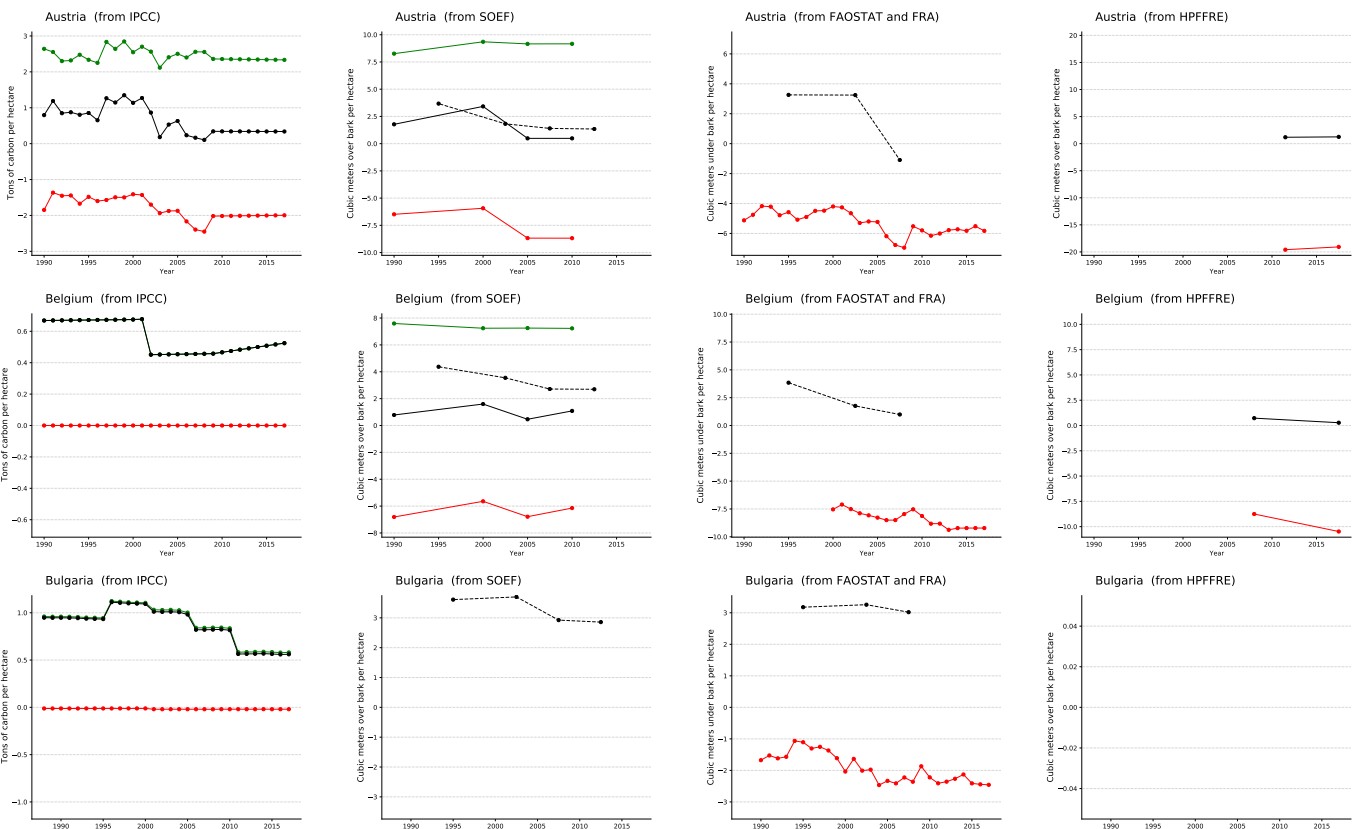

**Figure A1.** *Cont.*

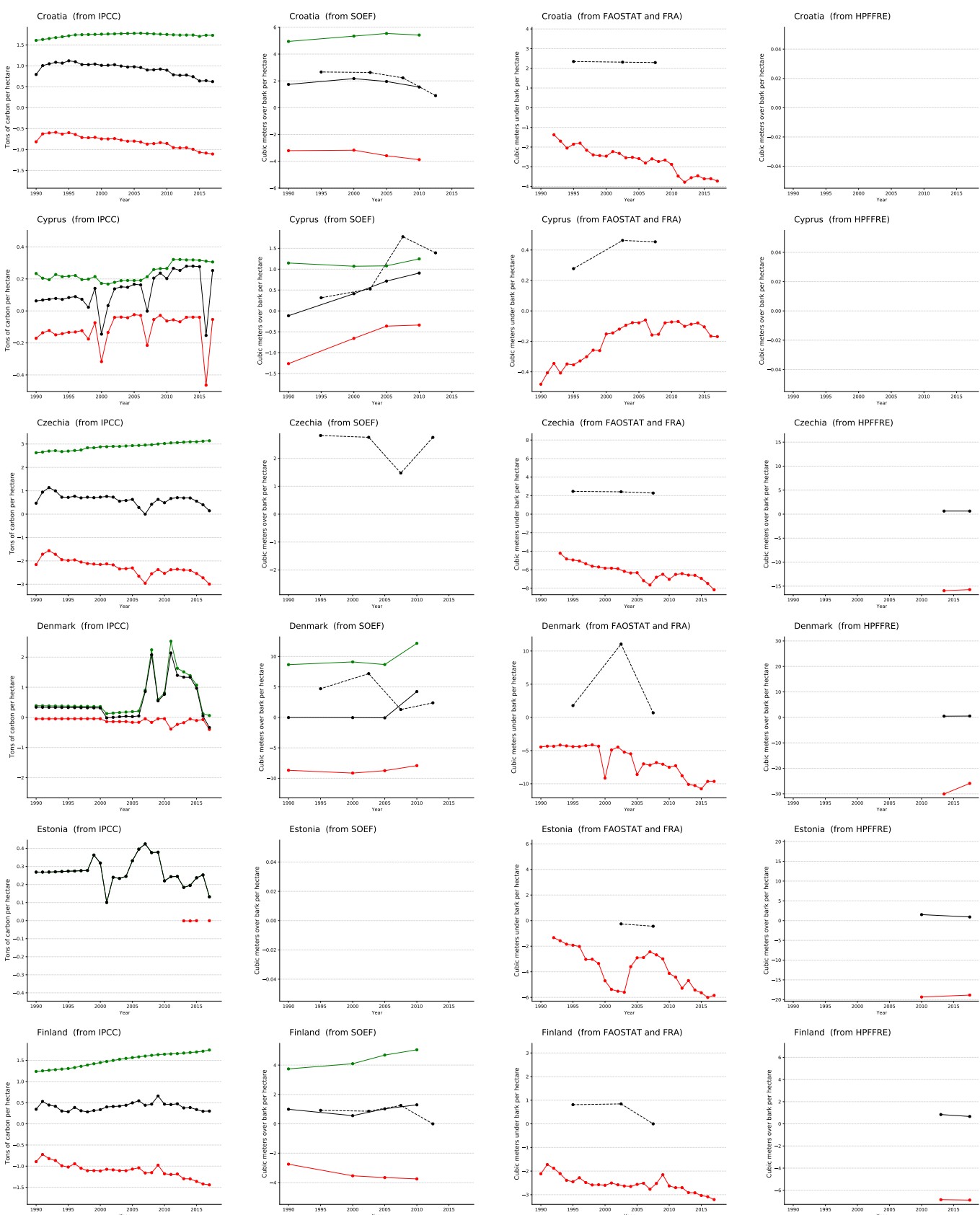

**Figure A1.** *Cont.*

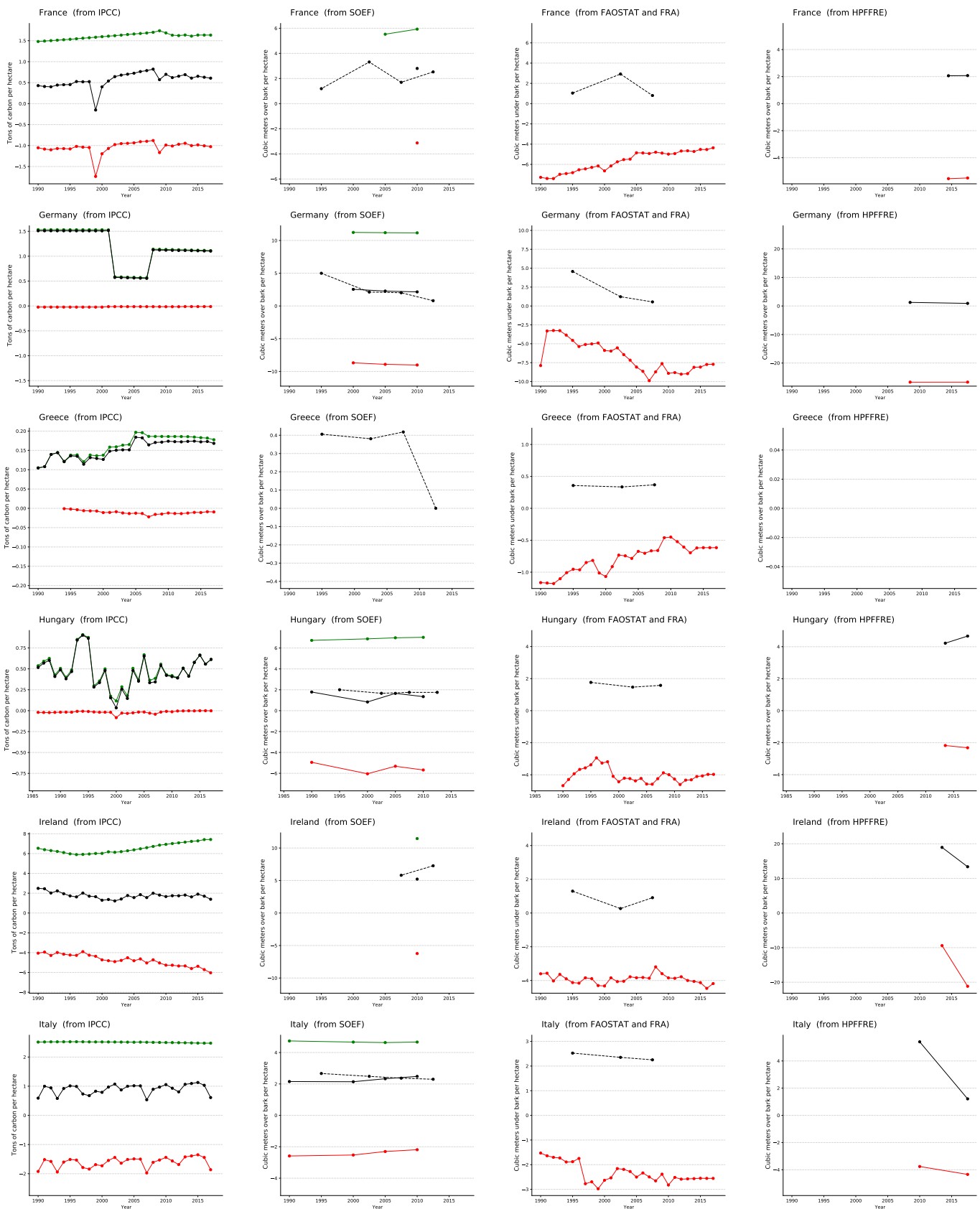

**Figure A1.** *Cont.*

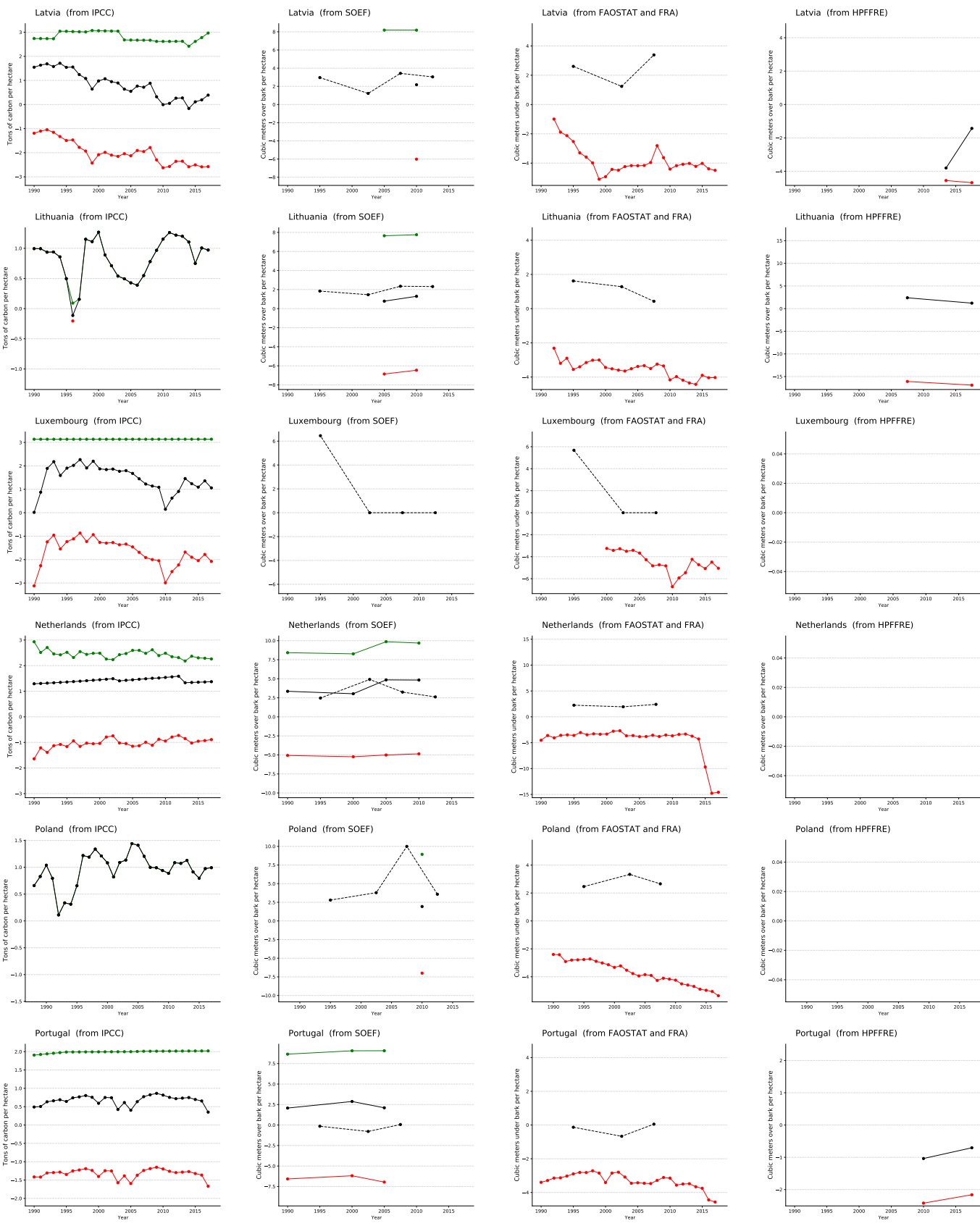

**Figure A1.** *Cont.*

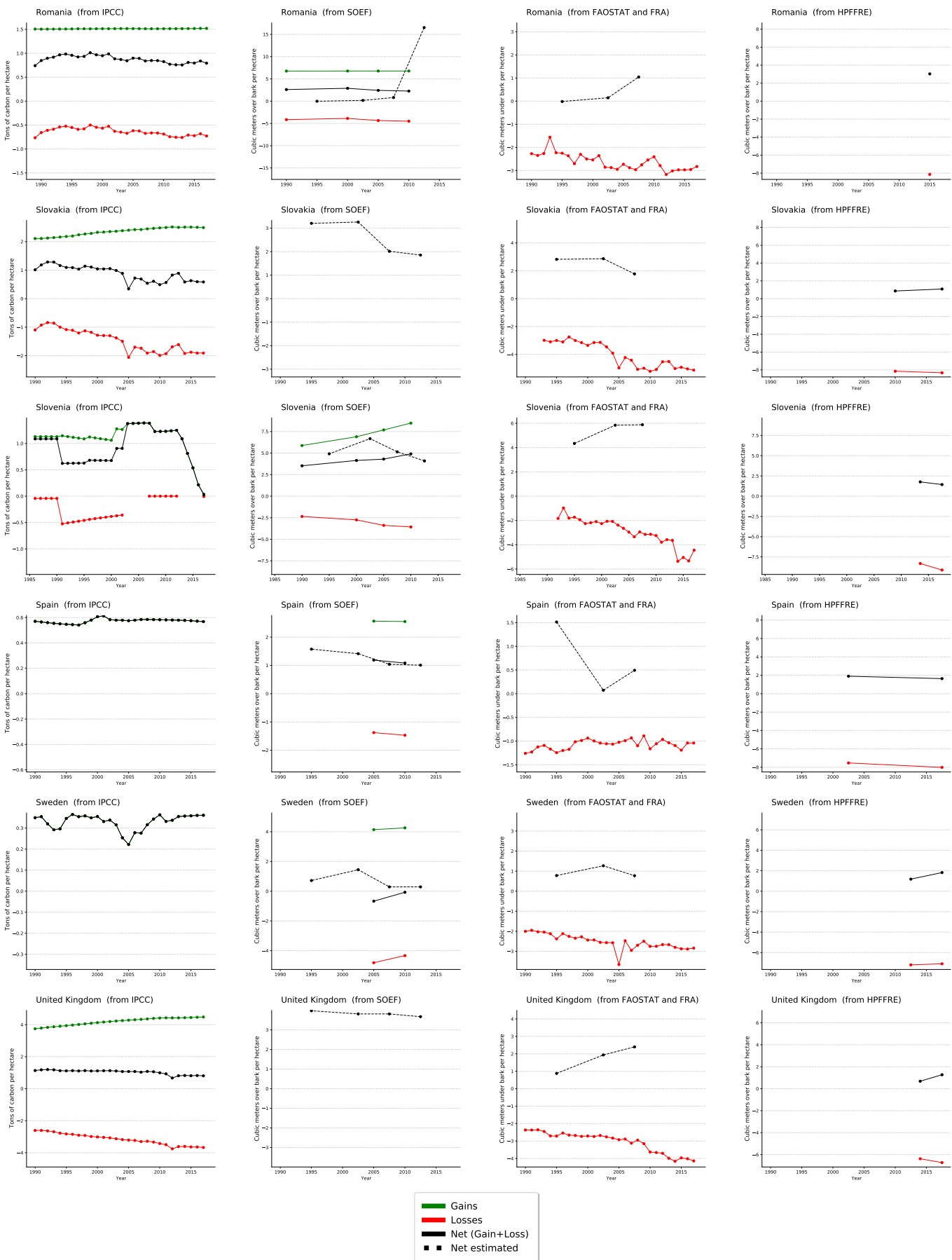

**Figure A1.** Comparison of the forest dynamics expressed in the original units for all countries.

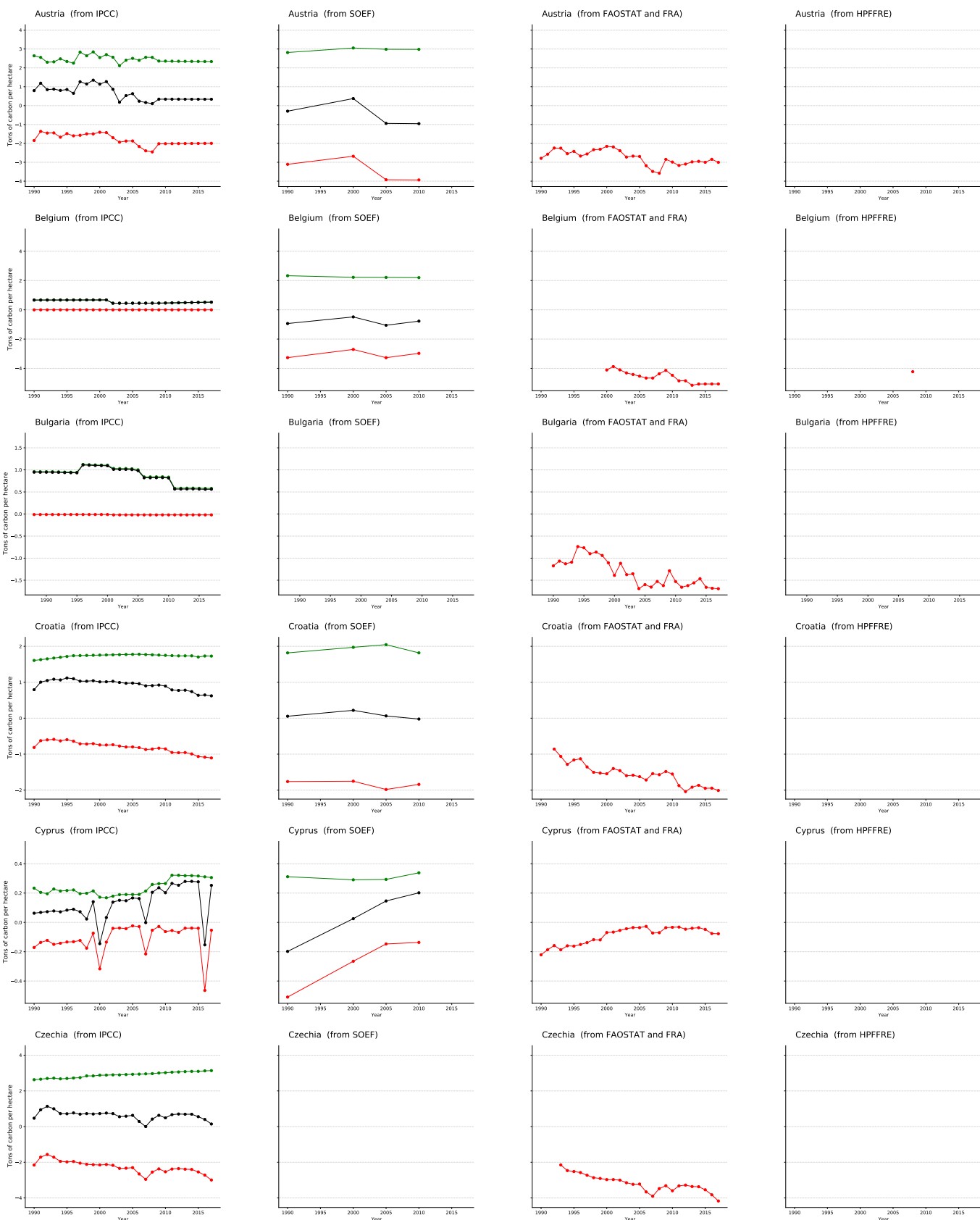

**Figure A2.** *Cont.*

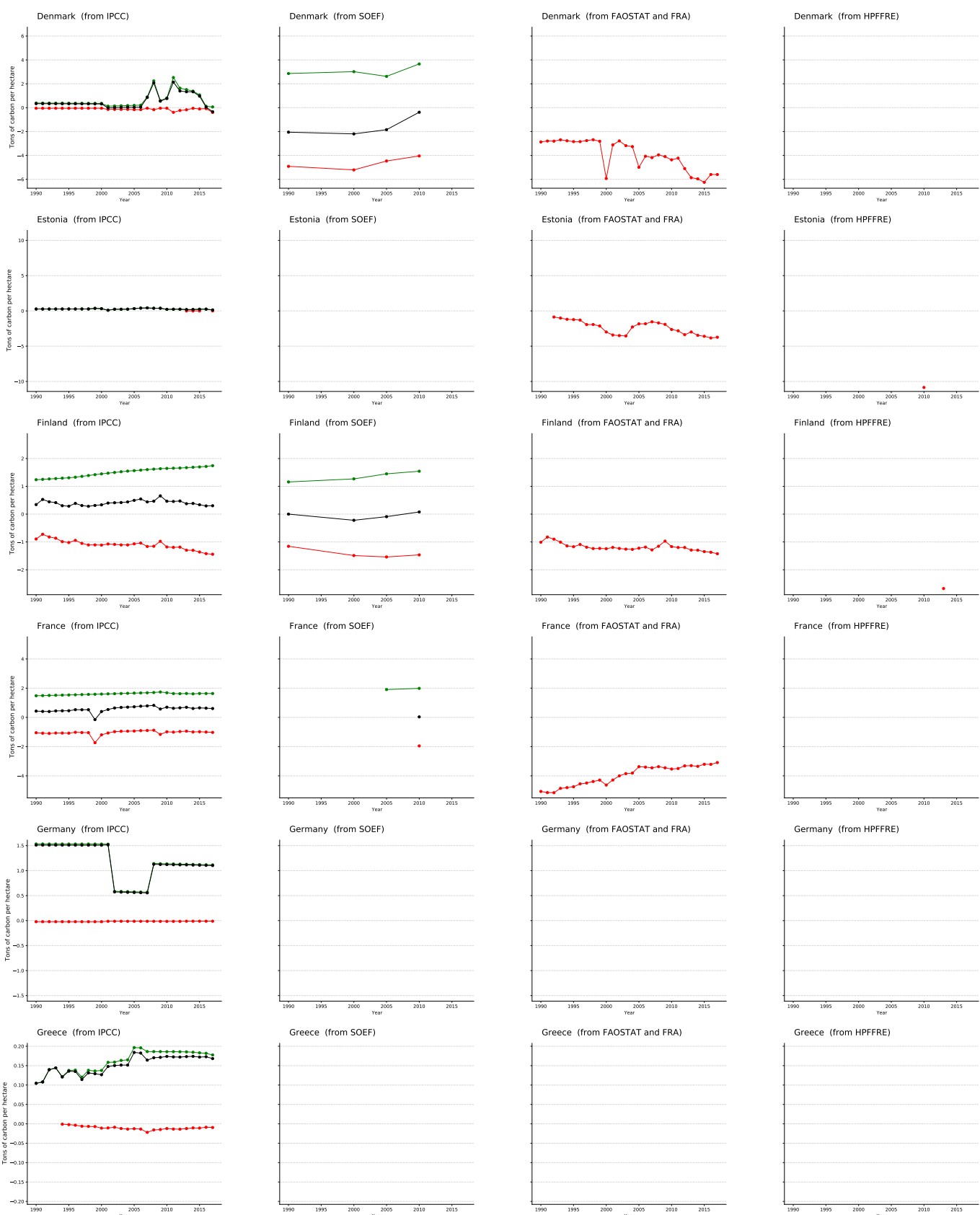

**Figure A2.** *Cont.*

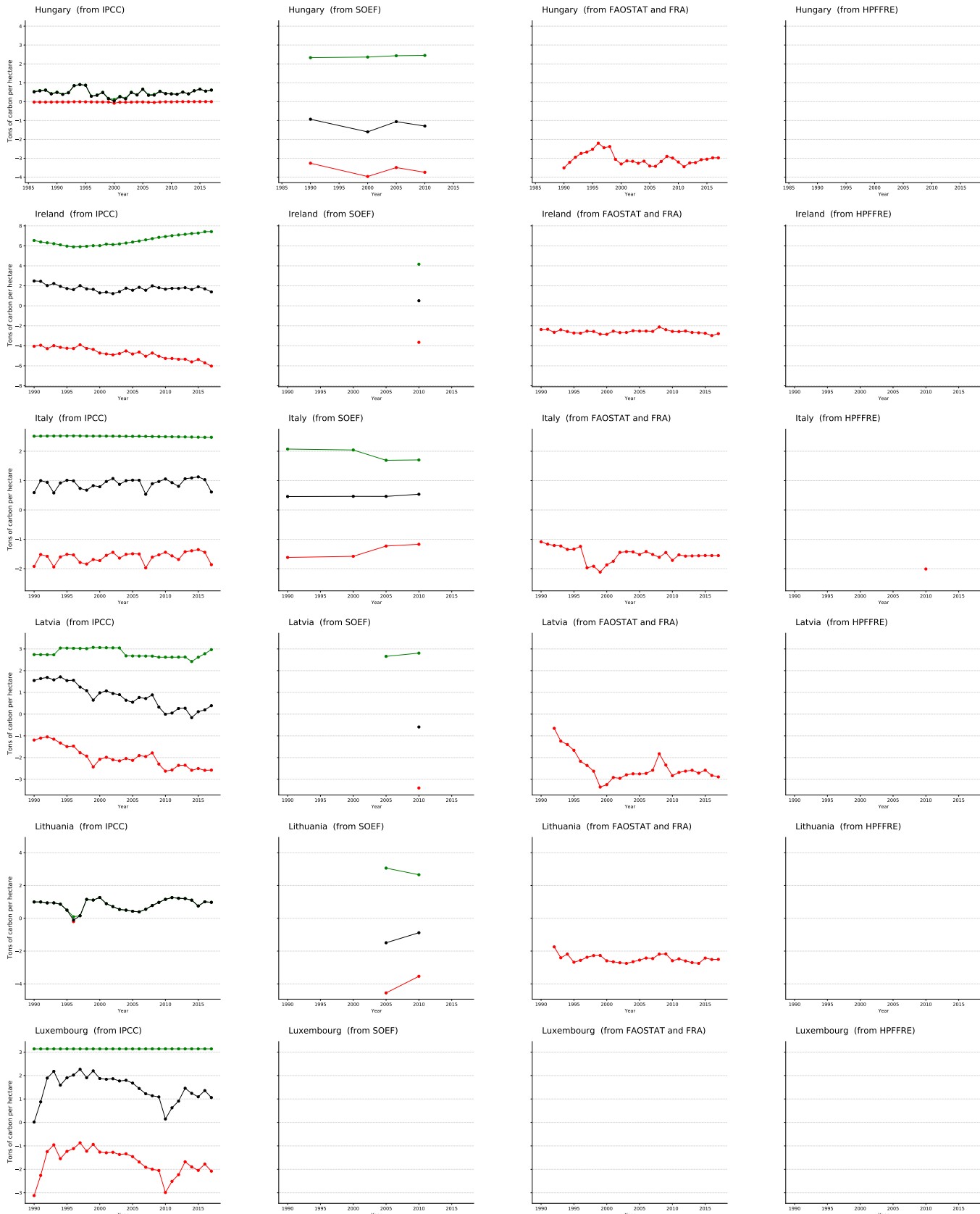

**Figure A2.** *Cont.*

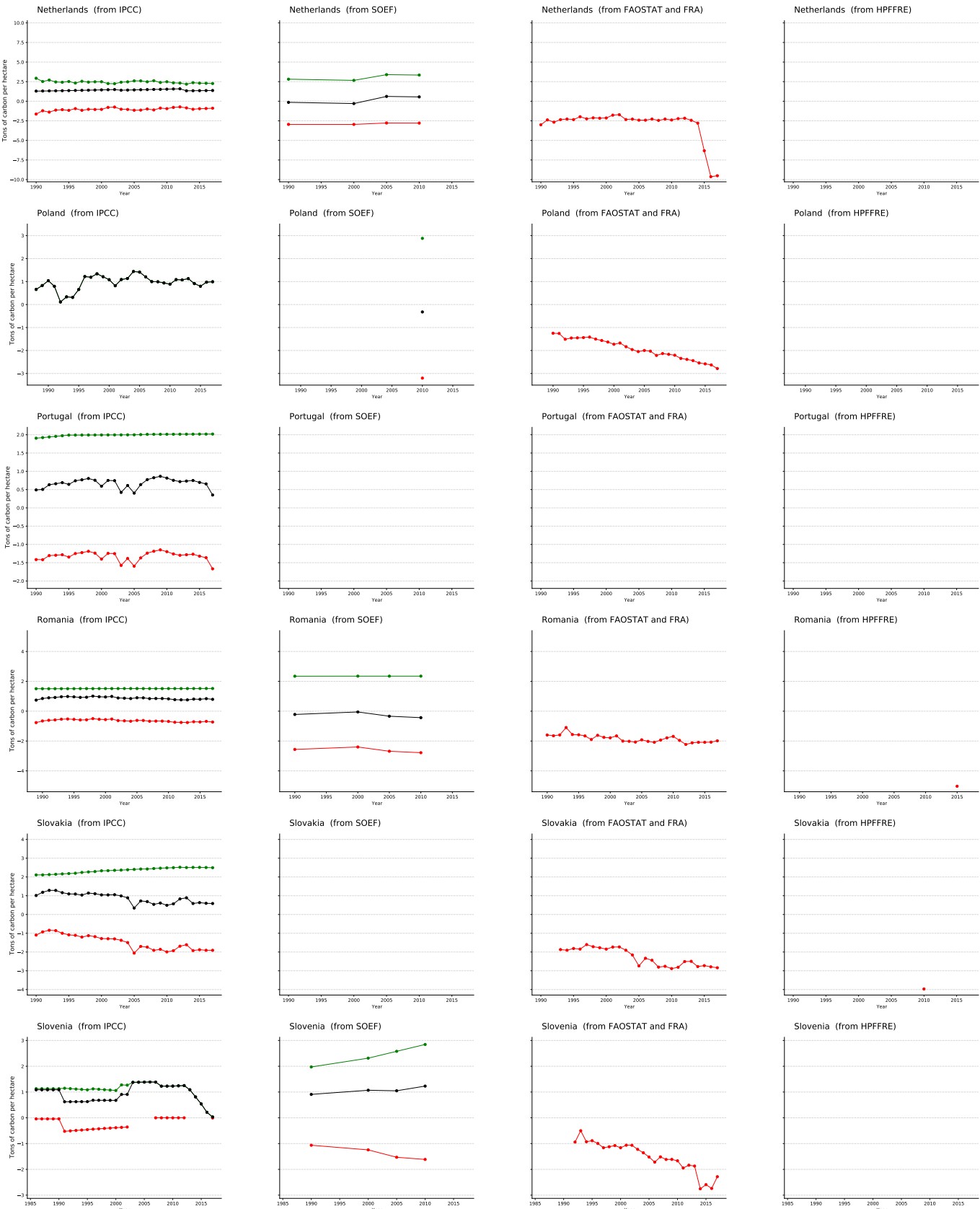

**Figure A2.** *Cont.*

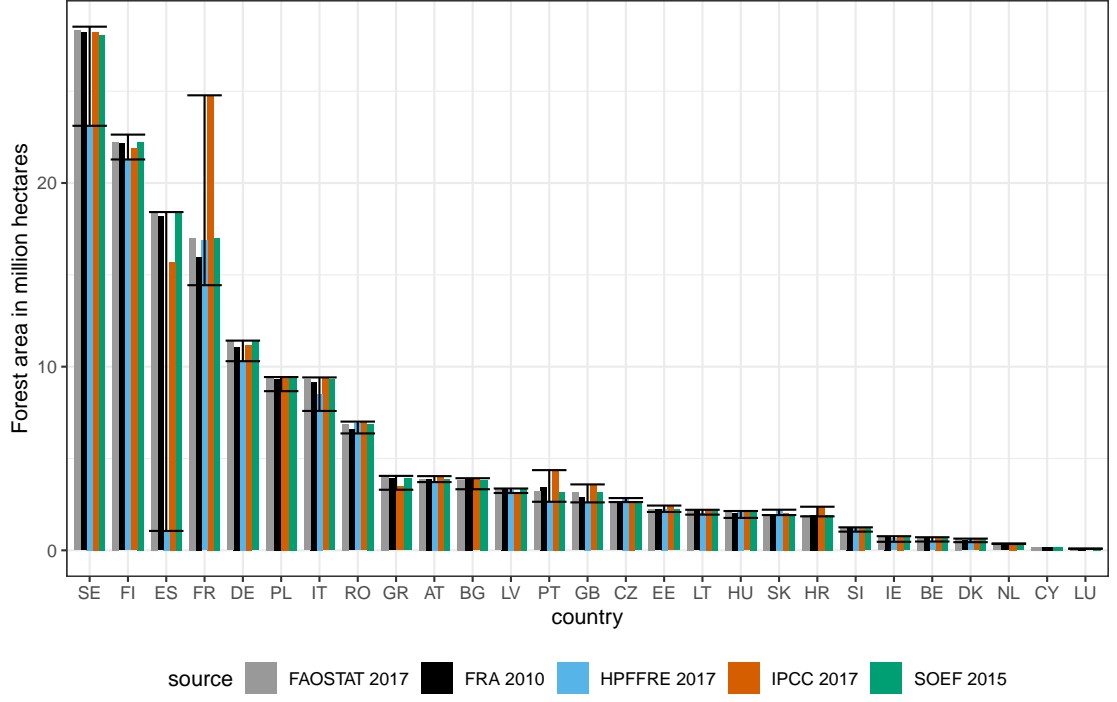

**Figure A2.** Comparison of the forest dynamics expressed in tonnes of carbon for all countries.

**Figure A3.** Forest area for the most recent year available in each dataset and each country. For each country, the error bars represent the minimum and maximum values over all time series and all sources.

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
