# Peer review of "Comparing Reported Forest Biomass Gains and Losses in European and Global Datasets"

_forests, doi:10.3390/f12020176_

Round 1
Reviewer 1 Report
lines 12-15, This conclusion is not new and it is reflected in multiple research projects providing relatively high correlation with national reports and forecasts as far as experts experienced in national NFIs are involved, e.g. implementation of EFDM model in the Diabolo project.
lines 36-38, Interpretation of NFI plots differs from country to country, starting from temporal of fixed plots, and finalizing with the internal variations within a plot, when it is split into sectors. It is also important to point in the introduction that national land use definitions may differ from the IPCC reporting requirements, e.g. criteria for forest lands, therefore NFI may not directly provide information on land use and carbon stock change in a format comparable with IPCC definitions.
lines 63-67, spatial data sources are missing in the methodology, e.g. Global forest watch is not very accurate source of information, but at EU level it can give insight into accuracy and trends in activity data. It should be mentioned in the description. The analyzed data sources lacks spatial data sources, which can provide valuable information on land use changes related carbon stock changes.
lines 80-81, it should be explained he how double accounting is avoided, because natural disturbances usually are later reported as harvests in countries with functioning forest management. It is also important to mention here how the natural mortality is treated, otherwise this potentially large source of GHG emissions is overlooked.
lines 86-87, It is important to add in the definition - living trees at the beginning of the period.
lines 119-128, it should be mentioned here also that LiDAR technologies provides opportunity to detect very accurate tree height and even changes if repeated measurements are available, e.g. in Estonia. Depending from LiDAR technology even individual trees can be detected and groups od species can be identified providing opportunity to use NFI as reference plots for interpolation of high resolution and very accurate forest maps. This information should also be reflected here.
lines 153-154, why such a huge amount of work is done instead of using freely available aggregation tools, e.g. https://rt.unfccc.int/locator? That would save a lot of time and give more accuracy by elimination of copy-paste mistakes. This should be explained in the article.
lines 161-162, As mentioned above Locator tool can be used to retrieve data. This sentence is not fully correct in terms of availability of the data retrieval system in the IPCC.
chapters 2.1.2-2.1.5, for other data sources than for IPCC potential differences in land use and stock, e.g. harvest stock definitions should be reflected in the article to avoid misinterpretation of merchantable wood and stemwood losses due to harvest. This is partly reflected in the results section, but short reflection in the methodology would increase the readability of the article.
lines 279-281, IPPC default BEFs should be used according to climate zones provided in the IPCC 2006 guidelines to conform to the requirements for tier 1 methodology. It should be pointed in the article if the climate zones used in the article conforms to the climate zones in the IPCC 2006, otherwise the estimates should be recalculated using tier 1 method according to IPCC 2006. It should be also explained more in details how climate zones are applied in recalculation to above-ground and below-ground biomass.
lines 285-286, according to equation 3.2.4 IPCC 2006 default value for carbon fraction for increment is 50%. The same for carbon losses (equation 3.2.7) and other equations in the guidelines. Use of 47% clearly leads to underestimation of carbon stock changes, even according to Table 4.3 of IPCC 2006. Considering that climate region specific values are used for BEFs, it is surprising that C fraction is not used in the same way. This should be recalculated in the article.
lines 317-318, it important to point in the article that forest infrastructure, e.g. ditch network, forest roads, nurseries etc. can be reported under forest land or settlements having significant impact on land use and land use change information.
lines 345-346, it is not always a case in the NFI or definitions of merchantable timber should be clarified, usually NFI counts timber with bark, and merchantable timber is usually accounted without bark and without firewood fraction. This should be clarified in the article.
lines 355-356, listing of countries is not correct, e.g. in Latvia significant decrease of net increment is observed since 1990. This should be corrected in the article.
Table 3, average values are quite useless considering that they could change significantly during time. The most recent values, e.g. 5 years average would be more informative. Historical data can be expressed as trend.
line 378, the term "natural disturbances" here is misleading if the definitions of IPCC are applied, because it excludes natural mortality, which doesn't fit under natural disturbances.
line 387-388, the conclusion on uncertainty is not correct for all EU, for the NFI there is no difference between the harvest types. It should be noted here that representation of felling types differs from country to country in their NFIs. Basically all losses are rather uncertain.
line 411-413, can't agree to this conclusion, the basis of NFI is hidden location of plots to avoid purposeful management and cheating. Other important component of the NFI is methodology ensuring long-term data comparability. More players calculating something would introduce more mess and would definitely disgrace the NFI system. Therefore this conclusion is not substantiated.
Author Response
Thank you for your thoughtful comments.
> lines 12-15, This conclusion is not new and it is reflected in multiple research projects providing relatively high correlation with national reports and forecasts as far as experts experienced in national NFIs are involved, e.g. implementation of EFDM model in the Diabolo project.
It is true that the data availability issues are not new. The only novelty that we
provide in this article is that we move the focus from static stock values, to the
dynamic, time series aspect of gains (growth) and losses (disturbances). We explain that
data availability issues continue to apply for those variables as well. To calibrate a
pan-European forest dynamics model from similar open data sets, gains and losses data
would need to be at a higher level of disaggregation in terms of forest types and
climatic conditions.
> lines 36-38, Interpretation of NFI plots differs from country to country, starting from temporal of fixed plots, and finalizing with the internal variations within a plot, when it is split into sectors. It is also important to point in the introduction that national land use definitions may differ from the IPCC reporting requirements, e.g. criteria for forest lands, therefore NFI may not directly provide information on land use and carbon stock change in a format comparable with IPCC definitions.
We specified that differences in land use change definitions can render carbon stock
changes figures incomparable between NFI reporting and IPCC reporting. But we also note
that "As countries have close to 90% of their forest area available for wood supply (see
table A1), it makes sense to compare biomass dynamics across these sources even though
they have a different forest definition."
> lines 63-67, spatial data sources are missing in the methodology, e.g. Global forest watch is not very accurate source of information, but at EU level it can give insight into accuracy and trends in activity data. It should be mentioned in the description. The analyzed data sources lacks spatial data sources, which can provide valuable information on land use changes related carbon stock changes.
Spatial data sources do provide valuable in formation on land use change, however the
article focuses on biomass dynamics in forest land remaining forest land for which it is
currently much harder to obtain biomass change measurements from remote sensing. We
mention spatial data sources in the paragraph of the introduction starting with
"advances in remote sensing". We also specify that "remote sensing based detection of
tree growth or small scale biomass loss remains highly uncertain" with a citation to
"Current situation and needs of change detection techniques" by Lu, Dengsheng and Li,
Guiying and Moran, Emilio. The high difficulty of change detection in forest land
remaining forest land was also the consensus emerging from a recent ESA workshop on
biomass change detection from remote sensing.
> lines 80-81, it should be explained he how double accounting is avoided, because natural disturbances usually are later reported as harvests in countries with functioning forest management. It is also important to mention here how the natural mortality is treated, otherwise this potentially large source of GHG emissions is overlooked.
Thank you, we added natural mortality to the losses. We specified that losses data
collected from the sources above do not distinguish between natural and anthropogenic
disturbances and that "The distinction is complex because changes in natural mortality,
anthropogenic and natural disturbances are frequently combined". The question of salvage
logging is also treated in the section results and discussion / disturbance dynamics.
> lines 86-87, It is important to add in the definition - living trees at the beginning
> of the period.
The definition was a quote from the COST action usewood. Another reviewer suggested to
remove the lengthy quote and simply refer to the citation for more details.
> lines 119-128, it should be mentioned here also that LiDAR technologies provides opportunity to detect very accurate tree height and even changes if repeated measurements are available, e.g. in Estonia. Depending from LiDAR technology even individual trees can be detected and groups od species can be identified providing opportunity to use NFI as reference plots for interpolation of high resolution and very accurate forest maps. This information should also be reflected here.
We mentioned the importance of LIDAR technologies in providing additional biomass
measurements to scale up field plot data to the level of satellite based measurements,
as explained in a review paper. "The importance of consistent global forest aboveground
biomass product validation" by Duncanson et al.
> lines 153-154, why such a huge amount of work is done instead of using freely available aggregation tools, e.g. https://rt.unfccc.int/locator? That would save a lot of time and give more accuracy by elimination of copy-paste mistakes. This should be explained in the article.
Thank you for the suggestion, we were not aware of this platform which can be very
useful. We had a look at the tool https://rt.unfccc.int/locator unfortunately it doesn't
distinguish emissions by losses and gains, which is the information we were trying to
obtain. For example countries latest submission for the sector 4.A.1 "Forest Land
Remaining Forest Land" contain time series of CO2 emissions but there are no data on
losses and gains.
> lines 161-162, As mentioned above Locator tool can be used to retrieve data. This sentence is not fully correct in terms of availability of the data retrieval system in the IPCC.
We explained first that the locator tool "does not distinguish forest land emissions by
losses and gain". And then we proceeded with the explanation of the painful download and
parsing process.
> chapters 2.1.2-2.1.5, for other data sources than for IPCC potential differences in
> land use and stock, e.g. harvest stock definitions should be reflected in the article
> to avoid misinterpretation of merchantable wood and stemwood losses due to harvest.
> This is partly reflected in the results section, but short reflection in the
> methodology would increase the readability of the article.
Under SOEF, FAO and HPFFRE we mentioned the differences in terms of land use and stock
definitions referring to the methodology section on the conversion to mass for more
details.
> lines 279-281, IPPC default BEFs should be used according to climate zones provided in
> the IPCC 2006 guidelines to conform to the requirements for tier 1 methodology. It
> should be pointed in the article if the climate zones used in the article conforms to
> the climate zones in the IPCC 2006, otherwise the estimates should be recalculated
> using tier 1 method according to IPCC 2006. It should be also explained more in
> details how climate zones are applied in recalculation to above-ground and
> below-ground biomass.
Thank you, we improved the explanation of how we implemented the choice of BCEF_I,
BCEF_R and R for each country, according to the climatic zones, forest type and
thresholds of biomass stock.
For information, the choice of conversion factors is documented in the source code at:
https://github.com/xapple/forest_puller/blob/master/forest_puller/conversion/bcef_by_country.py#L31
The allocation of each country to one or 2 ecological zones is stored in this reference
table:
https://github.com/xapple/forest_puller/blob/master/forest_puller/extra_data/country_codes.csv
Example instructions to load the input BCEF in python 3:
from forest_puller.conversion.load_expansion_factor import bcef_coefs
Example instruction to load the BCEF chosen for each country:
from forest_puller.conversion.bcef_by_country import country_bcef
> lines 285-286, according to equation 3.2.4 IPCC 2006 default value for carbon fraction for increment is 50%. The same for carbon losses (equation 3.2.7) and other equations in the guide> lines. Use of 47% clearly leads to underestimation of carbon stock changes, even according to Table 4.3 of IPCC 2006. Considering that climate region specific values are used for BEFs, it is surprising that C fraction is not used in the same way. This should be recalculated in the article.
We checked the IPCC 2006 guidelines, there are no equations in chapter 3 and no equation
3.2.4. The Carbon Fraction CF value that we used comes from chapter 4 under the
calculation steps for Tier 1.
https://www.ipcc-nggip.iges.or.jp/public/2006gl/pdf/4_Volume4/V4_04_Ch4_Forest_Land.pdf
carbon_fraction = 0.47
> lines 317-318, it important to point in the article that forest infrastructure, e.g. ditch network, forest roads, nurseries etc. can be reported under forest land or settlements having significant impact on land use and land use change information.
We specified that forest infrastructure can be reported under forest land. We do not
have a way to address this issue at the national scale. This issue is probably addressed
at the landscape level or at the plot level.
> lines 345-346, it is not always a case in the NFI or definitions of merchantable timber should be clarified, usually NFI counts timber with bark, and merchantable timber is usually accounted without bark and without firewood fraction. This should be clarified in the article.
The purpose of this transition sentence was to make another reference to the conversion
from m3 of merchantable timber, including bark, to tons of carbon. All forest dynamics
variables reported to international processes come from the NFI originally, but the
mention of NFI in this transition sentence may be misleading so we removed it.
> lines 355-356, listing of countries is not correct, e.g. in Latvia significant decrease of net increment is observed since 1990. This should be corrected in the article.
We added a sentence for Latvia which has decreasing gain values.
> Table 3, average values are quite useless considering that they could change
>significantly during time. The most recent values, e.g. 5 years average would be more
>informative. Historical data can be expressed as trend.
We added a figure that provide an overview of change over time, using error bars to show
the maximum and minimum values of the gains and losses over the time series. That figure
is expressed in original unit, while the table of average gains and losses is converted
to tons of carbons. The figure and table are complementary, therefore we left the table
in the article. The table also offers a comparison points to which we refer in the text.
> line 378, the term "natural disturbances" here is misleading if the definitions of IPCC are applied, because it excludes natural mortality, which doesn't fit under natural disturbances.
We added that "anthropogenic carbon emissions lead to increases in natural disturbances
and mortality"
> line 387-388, the conclusion on uncertainty is not correct for all EU, for the NFI there is no difference between the harvest types. It should be noted here that representation of felling types differs from country to country in their NFIs. Basically all losses are rather uncertain.
We removed the statement on uncertainty being higher for natural disturbance without
salvage logging. We replaced it by a more general statement valid for all EU countries:
"in the international sources analysed in this article, disturbances loss estimates are
reported without any indication of the associated uncertainty".
> line 411-413, can't agree to this conclusion, the basis of NFI is hidden location of
plots to avoid purposeful management and cheating. Other important component of the NFI
is methodology ensuring long-term data comparability. More players calculating something
would introduce more mess and would definitely disgrace the NFI system. Therefore this
conclusion is not substantiated.
We replaced "secret" by "statistical confidentiality" and explained that over the long
term, a form of harmonised public release of information should aim to get "as close as
possible to these primordial quantifications, while continuing to ensure statistical
confidentiality."
On the question of more players calculating something, we hope that what we mean by
harmonisation is not misunderstood. International processes are another level on top of
NFI internal data collection mechanisms. We wish there would be improvements to the data
provided by international processes that could be used in a forest dynamics model at the
European scale. As we state in the conclusion upcoming satellite sensors will make high
resolution biomass maps available on a regular basis. In the next decades, concerted
release of NFI data on biomass losses and gains are probably going to be needed to help
calibrate those maps. Such publicly available, harmonized ground data (protected by
aggregation mechanisms ensuring confidentiality) can also provide invaluable data for
forest dynamics modelling.
Reviewer 2 Report
Authors compare different values from various forest databases in EU27 countries. The paper is not well written, it is more like a novel rather a scientific text. The graphs are hard to read and are too many bearing a small value of information. The text has many times incorrect English. Citations are placed into the text in very inapropriate way. There is very small of discussion in results part and current Discussion section is more like Conclusions. Authors should be careful with the surface data information, as in some countries National parks are not included in those databases.
Here are more detailed comments.
line 38-40. Stating a reason why not with explanation that it is out of the scope of the paper is somehow weird. Put (iii) out.
line 48: related to what? Please rewrite the sentence
line 50: "Now coming back to our analysis" - it is not a speech but scientific text. Such sentences are higly inapropriate.
line 51-51: "provided the source in question includes a similarly formatted dataset for 51 every one of the 27 EU member states" - the sentence does not make sense
line 52: data are plural. Therefore there should be "originate". The same in line 56, 59,111... Check it throughout the manuscript.
line 58: Could you describe the estimation in more detail?
line 81: include also tropospheric ozone and other pollutants
line 86-91. Delete this. It is not common to cite such large text in science. Put only reference.
Figure 1: Do not write "This diagram shows", rather write what it shows.
line 92: "As the figure shows" - maybe it seems it is clear that you mean Figure 1. It is much better, however, to write, for example: "Net increment is the gross increment minus natural losses (Fig. 1). Making the text longer that is needed without information is not how to write scientific text.
line 98-100. Do not use "In another words", it is not spoken english. Anyway, it seems that you are repeating what you wrote above. Why to double the information? If you are not sure that the reader understands, than rewrite the paragraph to make it clear. The same in line 127.
line 102: why do you first in text link to Figure A1, when there was not previous link to Table 1?
line 107-108: why is the text in quotation marks?
line 142-145. Do you think that reader does not know member states of the EU? Delete the countries, this is common information who is the member.
lines 153-241. Needs to be completely rewritten. Noone is interested if the data download was hard, as you write or if it was in .csv or Excel or that each database has the same variable in different row. Make one sentence from those lines, for example: "The data were downloaded from five databases". Moreover,. never put link reference to the text. You have references at the end of manuscript!
line 255: I see four sources having stem wood volume.
line 258: why do you write here Eq. 2.10, when in here it is Eq. (1)? Do not write here "based on equation 2.10 of the IPCC guidelines", but only Eq. (1) and reference. Do not write here that is is in chapter two! That could be in references too! The same for line 269.
line 260: "Where" should not be in capitals. You continue the sentence.
lines 261-267. do not write "expressed in", just define the variable and the unit put in brackets. The same for lines 271-275.
line 264: do not write "The unis are"
line 268: you define removal volume as Hv, do not write that it is H as harvest.
line 276: Do not write here about table 4.4, report only the reference and the reader could find the exact table by himself. As now it only brings confusion that you have table 4.4 in here. Moreover, write only the refence, why do you write in text .
line 280-281. The link should be in references.
line 285: chapter 4 put into the references
line 288: do not write "taken from [18, page 19]", just put the reference! Moreover, the page should be in references at the end of manuscript, not here.
line 200: what is the hypothesis and how you simplify that? Moreover, you state, that it should not be used for exact comparison. But what you do in this manuscript? You do exact comparison.
lines 203-311: do not write that it was easier. You describe the proces, not your emotions. "avoiding any GUI or dependency on commercial Microsoft technologies" - inapropriate - delete this.
Of course reader assumes that the equations here are also in the code! Do not write that it is there, we expect that. All this section needs to be rewritten, it has very limited amount of information and is too long with vague sentences. It is not a novel, but scientific text.
line 300: what is this? [? ]
line 313-314: delete this.
line 217: It is not novel! "it is natural to start comparing forest areas first" - so just start to compare, do not write such sentences which do not bring any information. Go directly to the point.
Figure 2: I cannot see the axis description, it is too small, make it readable. Y axis description should be: "Area (ha * 10E6)". Write description and units in brackets, not all as text.
Table 2: what are the units of forest area? How value like 4’040’500 could be one number representing surface area?
Figure 2,3,4: Delete those figures. It holds only very little information. It would be much better to show only the differences between the databases in different global graphs or making one table. Having three pages of those graphs is not a good way.
line 380: which countries, be specific.
line 383-384. Much more hit by stomrms was Slovakia? Do you see striking illustration also there? Do you see huge bark beatle outbrake in Czechia?
line 389: "moving further" should not be in scientific text.
line 390-392: I think, that carbon emissions are not in scope of your paper, so why do you write baout that here? Notabyly when the effect is not separtable.
Discussion: It is not discussion, it is Conclusion. Discussion part is therefore missing, however is present in Results. The results part must be rewritten and the discussions there should be places into teh Discussion part. Current Discussion part will become Conclusions.
Author Response
Thank you for your helpful comments.
> line 38-40. Stating a reason why not with explanation that it is out of the scope of
> the paper is somehow weird. Put (iii) * out.
Removed point (iii).
> line 48: related to what? Please rewrite the sentence
Added "related to this framework contract"
> line 50: "Now coming back to our analysis" - it is not a speech but scientific text. Such sentences are higly inapropriate.
Replaced by "In the following analysis".
> line 51-51: "provided the source in question includes a similarly formatted dataset for 51 every one of the 27 EU member states" - the sentence does not make sense
We replaced by a synonym "on condition that"
> line 52: data are plural. Therefore there should be "originate". The same in line 56, 59,111... Check it throughout the manuscript.
We corrected verbs to the plural form at the following places:
* All the data acquired by these external sources [-originates-]{+originate+} at the individual NFIs
* The data [-is-]{+are+} summarized and filtered to
* The data that we examine in our study [-is-]{+are+} the outcome of
* Inventory data [-is-]{+are+} scarce with a periodicity of 5 to 10 years
* Every country's data [-was-]{+were+} contained in a single Excel file.
* The data for wood removals [-is-]{+are+} downloaded
* FAOSTAT bulk data [-is-]{+are+} much easier to retrieve
* More data [-is-]{+are+} collected on harvest volumes
> line 58: Could you describe the estimation in more detail?
We added this sentence: "To expand data, national correspondents can use regression
techniques when data is available from case studies within the country or in other
countries within the same biome."
> line 81: include also tropospheric ozone and other pollutants
We added "atmospheric pollutants" as a generic anthropogenic disturbance.
> line 86-91. Delete this. It is not common to cite such large text in science. Put only reference.
We remove the lengthy definition and kept the citation and reference.
> Figure 1: Do not write "This diagram shows", rather write what it shows.
We shortened the figure description accordingly.
> line 92: "As the figure shows" - maybe it seems it is clear that you mean Figure 1. It is much better, however, to write, for example: "Net increment is the gross increment minus natural losses (Fig. 1). Making the text longer that is needed without information is not how to write scientific text.
We shortened the sentence and removed the definition, leaving only the reference to Vidal2016.
> line 98-100. Do not use "In another words", it is not spoken english. Anyway, it seems that you are repeating what you wrote above. Why to double the information? If you are not sure that the reader understands, than rewrite the paragraph to make it clear.
Thank you, we removed the duplicated content. The paragraph is hopefully clearer.
> The same in line 127.
Replaced "in other words" by "therefore" as it is a consequence of the previous sentence.
> line 102: why do you first in text link to Figure A1, when there was not previous link to Table 1?
We think table A1 is not very important and should remain an annex. Annexes receive a
number starting with "A".
> line 107-108: why is the text in quotation marks?
Because it is a citation. We could paraphrase it, but it is clearer in the quoted form.
> line 142-145. Do you think that reader does not know member states of the EU? Delete the countries, this is common information who is the member.
We removed the list of states.
> lines 153-241. Needs to be completely rewritten. Noone is interested if the data download was hard, as you write or if it was in .csv or Excel or that each database has the same variable in different row. Make one sentence from those lines, for example: "The data were downloaded from five databases". Moreover,. never put link reference to the text. You have references at the end of manuscript!
The methods section is indeed more technical and describes the procedures used to produce the software and the visualizations. This naturally might not of be of interest to as many readers. However, the inclusion of a detailed methods section showing the steps taken remains essential in a scientific publication, if only for reproducibility sake.
The reference links cannot be placed at the end of the manuscript unfortunately, as URLs are not authorized by the journal's formating rules (c.f. chicago2.bst). We thus choose to keep the URLs inside the text.
> line 255: I see four sources having stem wood volume.
Yes but the sentence is about forest dynamics (gains and losses) and the fourth source FRA only reports
about the stock, not the dynamics.
> line 258: why do you write here Eq. 2.10, when in here it is Eq. (1)? Do not write here "based on equation 2.10 of the IPCC guidelines", but only Eq. (1) and reference. Do not write here that is is in chapter two! That could be in references too!
> The same for line 269.
We placed the chapter number in the bibliographical references and simplified the equation number.
> line 260: "Where" should not be in capitals. You continue the sentence.
Updated.
> lines 261-267. do not write "expressed in", just define the variable and the unit put in brackets. The same for lines 271-275.
> line 264: do not write "The unis are"
We removed those small sentences and kept only the units in brackets.
> line 268: you define removal volume as Hv, do not write that it is H as harvest.
Removed that parenthesis.
> line 276: Do not write here about table 4.4, report only the reference and the reader could find the exact table by himself.* As now it only brings confusion that you have table 4.4 in here. Moreover, write only the refence, why do you write in text .
Removed the table number. Kept the table title as a pointer for the reader to find the exact table in the IPCC report.
> line 280-281. The link should be in references.
We placed all links into the bibliography www entries.
> line 285: chapter 4 put into the references
We removed that mention.
> line 288: do not write "taken from [18, page 19]", just put the reference! Moreover, the page should be in references at the end of manuscript, not here.
We updated the references.
> line 290: what is the hypothesis and how you simplify that? Moreover, you state, that it should not be used for exact comparison. But what you do in this manuscript? You do exact comparison.
We specified the simplifying hypothesis made in the choice of BCEF and R. We rephrased
the sentence on "exact comparison" to mention that there is a large margin of error.
> lines 203-311: do not write that it was easier. You describe the proces, not your emotions. "avoiding any GUI or dependency * on commercial Microsoft technologies" - inapropriate - delete this.
We rephrased to describe the reproducibility issue generated by free form spreadsheets.
We delete the inappropriate content.
> Of course reader assumes that the equations here are also in the code! Do not write that it is there, we expect that. All this section needs to be rewritten, it has very limited amount of information and is too long with vague sentences. It is not * a novel, but scientific text.
One aim of this technical paper is to encourage reuse of the software. We hope other
researchers will find the python software useful for their modelling activities and will
contribute improvements to the data preparation and transformation steps. Unfortunately
a large part of the code deals with uninteresting data download and formatting steps. By
showing where the main equations are implemented we help the reader enter into the
interesting part of the code base.
> line 300: what is this? [? ]
We replaced @software by @Article in the bibliography reference file and the reference now appears correctly.
> line 313-314: delete this.
We deleted the sentence.
> line 217: It is not novel! "it is natural to start comparing forest areas first" - so just start to compare, do not write such sentences which do not bring any information. Go directly to the point.
We agree with you on the need to go directly to the point and this is surely what an
expert would expect, but here we wanted to add a transition sentence so that the flow of
our reasoning can be understood more clearly. We wanted to emphasise that forest area is
not the central topic of the paper, in fact we only analyse the area because we use it
in the next step to normalise biomass losses and gains values.
> Figure 2: I cannot see the axis description, it is too small, make it readable. Y
axis description should be: "Area (ha 10E6)". Write description and units in brackets,
not all as text.
We increased the font size of the Y axis label.
> Table 2: what are the units of forest area? How value like 4’040’500 could be one number representing surface area?
We added the unit to the table title. The tick marks represent a thousand separator.
> Figure 2,3,4: Delete those figures. It holds only very little information. It would be much better to show only the differences between the databases in different global graphs or making one table. Having three pages of those graphs is not a * good way.
We replaced figure 3 by a smaller plot of average losses and gains by country. Thank you
for your comment. This new figure 3 provides a more concise overview of how the
different dataset compare to each other. However, several paragraphs refer to the trend
of the time series, both in terms of the area and of the evolution of losses and gains
over time. That is why we think it is important to keep the time series plot illustrated
figure 2 and 4. We added a summary of the forest area data as a supplementary figure A3
which illustrates the scale of area differences across sources and countries.
> line 380: which countries, be specific.
We added the countries in the text.
> line 383-384. Much more hit by stomrms was Slovakia? Do you see striking illustration also there? Do you see huge bark beatle * outbrake in Czechia?
In the annex figures A1 and A2, increases in harvest are visible in both the IPCC and
FAOSTAT data for Slovakia around 2005 and from 2014- onwards. The extend of the outbrake
might not have been fully reported by the country to those international processes yet.
> line 389: "moving further" should not be in scientific text.
We removed these words from the text.
> line 390-392: I think, that carbon emissions are not in scope of your paper, so why do you write baout that here? Notabyly when the effect is not separtable.
Because natural disturbances due to these emissions could have been treated in the IPCC
loss data in a different way and they could have been absent from the SOEF data. That
sentence specifies that natural disturbances due to anthropogenic emissions are not
separable. They are therefore treated in a similar way in all sources.
> Discussion: It is not discussion, it is Conclusion. Discussion part is therefore missing, however is present in Results. The results part must be rewritten and the discussions there should be places into teh Discussion part. Current Discussion part will become Conclusions.
We renamed the "discussion" section to conclusion and rename "results" to "results
and discussion".
Reviewer 3 Report
The paper compares different data sources in an international context. It is found that the results vary. It is concluded that the quality of the data is not good enough to allow EU-level scenarios.
The content of the paper is mainly technical, instead of scientific research. However, the Aims and Scope of the Journal contains a statement “we also accept manuscripts communicating to a broader audience”. This reviewer takes this as an indication of suitability to the Journal.
The reviewer has not found any major methodological issues from the paper. This partially reflects the technical character of the content.
The reviewer has some difficulty in figuring out what the results represent. Fig. 2 is clear. Do Figs. 3 and 4 show annual changes, instead of cumulative changes? Do I understand correctly that “gains” refers to growth, “losses” to harvesting?
Rather many tables are mentioned in the text. Are they available somewhere as supplementary material? I cannot find them now.
The content of Table A1 appears to require clarification.
Author Response
Thank you for your comments.
> The paper compares different data sources in an international context. It is found
> that the results vary. It is concluded that the quality of the data is not good enough to allow EU-level scenarios.
> The content of the paper is mainly technical, instead of scientific research. However,
> the Aims and Scope of the Journal contains a statement “we also accept manuscripts
> communicating to a broader audience”. This reviewer takes this as an indication of
> suitability to the Journal.
This is indeed a technical paper describing the software used in a preparatory step for
forest dynamics modelling. It is also a data paper, because the software tool
facilitates access to international reference data on forest dynamics: IPCC losses and
gains, FAOSTAT harvest, SOEF increments and fellings, HPFFRE harvest.
> The reviewer has not found any major methodological issues from the paper. This
> partially reflects the technical character of the content.
> The reviewer has some difficulty in figuring out what the results represent. Fig. 2 is
> clear. Do Figs. 3 and 4 show annual changes, instead of cumulative changes? Do I
> understand correctly that “gains” refers to growth, “losses” to harvesting?
Figures 3 has been replaced by another figure in the main part of the article. The
content of the original figure 3 is still visible in annex as figure A1. Figure 4, A1
and A2 all represent the annual value of losses and gains i.e. the annual changes of
biomass. Gains and losses refer to the IPCC terminology, while increments and fellings
refer to the SOEF terminology. We have improved the methodology section to explain how
we used biomass conversion and expansion factors and root to shoot ratios to convert the
increments and fellings expressed in m3 of wood over bark so that they can be compared
to IPCC gains and losses expressed in tons of carbon.
> Rather many tables are mentioned in the text. Are they available somewhere as
> supplementary material? I cannot find them now.
Tables 1, 2 and 3 are in the main body of the article and table A1 is in the annex.
All tables should now be visible in the pdf.
> The content of Table A1 appears to require clarification.
We added an explanation of what FRAWS mean (Forests with Restrictions on Availability
for Wood Supply) and we updated the sentence referring to that table.
Round 2
Reviewer 2 Report
Thank you for the changes you made. I think the manuscript is now significantly better.